# RetrOrchestrator: A Multi-Step Retrosynthesis Agent Dynamically Orchestrating Single-Step Transition Models

Chang Liao[1]  Luotian Yuan[2]  Yiping Ke, Kelly[1]  Ying Wei[2]

## Abstract

Multi-step retrosynthesis planning is a fundamental challenge in organic chemistry, defined by its enormous search space. Existing methods typically formulate it as a Markov Decision Process (MDP) with a fixed choice of transition model (i.e., a single-step retrosynthesis model), and focus on improving *how to search* through better policies and value functions. However, *how the transition space itself is navigated* remains largely unexplored. This limitation is particularly urgent given our observation of pronounced *skill disparity* among single-step prediction models: different models exhibit substantially different performance across molecule states. Motivated by this observation, we introduce RetrOrchestrator, an LLM-powered agent that explicitly accounts for model skill disparity by reframing retrosynthesis planning as a Partially Observable Markov Decision Process (POMDP). By regarding each single-step prediction model as a tool, we further propose a scaffold-aware reinforcement learning algorithm to optimize navigation policy within the transition space. As a result, RetrOrchestrator jointly searches which molecule to expand and which single-step model to apply for the molecule at the current step. Empirically, RetrOrchestrator significantly outperforms static baselines on the Retro*-190 benchmark, achieving a state-of-the-art 94.21% success rate (vs. 9.47% off-the-shelf LLM and 82.63% non-LLM state-dependent router), with 92.49% of solved routes invoking two or more SSRs—evidence that the policy is not collapsing to a single specialist or a static router. The same gain persists on a larger out-of-distribution set (PDB-600), with RetrOrchestrator

Pareto-optimal in both wall-clock time and model-query count. Code: `https://github.com/ScottLiao920/verl-retro-agent`.

## 1. Introduction

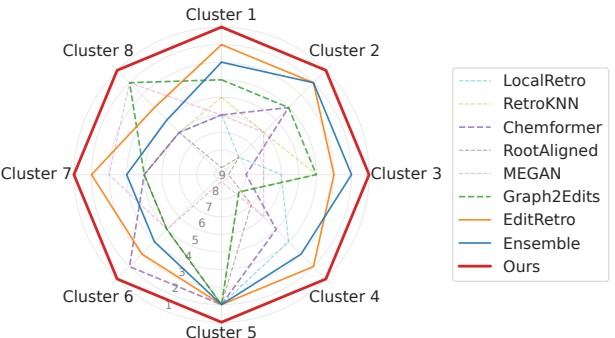

*Figure 1.* Relative performance of retrosynthesis agents with different single-step retrosynthesis models on the Retro*-190 benchmark, clustered by target molecule scaffold. The radial axis shows the relative ranking within each cluster, ranging from 1 (best) to 9 (worst). Molecules in each cluster are visualized in Appendix R.

Multi-step retrosynthesis planning is one of the most fundamental and challenging problems in organic chemistry: given a target molecule, the goal is to identify a sequence of chemical reactions that transforms readily available commercial molecules into the target. The space of possible synthetic routes is astronomically vast, with an estimated $3.5 \times 10^{28}$ possibilities for a single 15-step pathway (Szymkuć et al., 2016). Consequently, modern computational approaches (Segler et al., 2018; Chen et al., 2020; Schwaller et al., 2020; Genheden et al., 2020; Hong et al., 2023; Tu et al., 2025) emerge to navigate this combinatorial space, by framing retrosynthesis planning as a sequential decision-making process over a growing search tree.

At each step, a planner selects an intermediate molecule from the current retrosynthetic tree for expansion. This process can be naturally formulated as a Markov Decision Process (MDP), where the state represents the current search tree and the action selects which molecule to expand. Applying a reaction to the selected molecule induces a state transition, generating precursor molecules and updating the

[1]College of Computing and Data Science, Nanyang Technological University, Singapore [2]College of Computer Science and Technology, Zhejiang University, China. Correspondence to: Ying Wei <ying.wei@zju.edu.cn>.

*Proceedings of the 43rd International Conference on Machine Learning*, Seoul, South Korea. PMLR 306, 2026. Copyright 2026 by the author(s).

search tree accordingly. Central to this process are (1) *multi-step search algorithms* (Segler et al., 2018; Schreck et al., 2019; Chen et al., 2020; Yu et al., 2022; Yuan et al., 2024) determining which molecule in the current tree to expand, based on heuristics, learned value functions, or search policies, and (2) *single-step retrosynthesis models (SSR)* (Chen & Jung, 2021; Sacha et al., 2021; Irwin et al., 2022; Zhong et al., 2022; Xie et al., 2023; Zhong et al., 2023; Han et al., 2024) proposing candidate precursors to expand the selected molecule and defining the transition dynamics.

Under this MDP formulation, a critical limitation of existing frameworks is their reliance on a single, static SSR throughout the entire search process, regardless of the intermediate molecules encountered. This static choice of transition model prevents the planner from exploring higher-quality retrosynthetic pathways, since SSR models exhibit pronounced *skill disparity* across molecular states. As we empirically show in Figure 1, no single SSR model is universally optimal. In Cluster 6 which consists of complex fused-ring scaffolds with diverse side chains, a sequence-based SSR model named Chemformer (Irwin et al., 2022) strikingly and significantly outperforms both recent graph-based models and their ensemble, despite underperforming in other chemical spaces. Thus, we are motivated to ask,

*Can we move beyond static planning by dynamically selecting the most suitable SSR model for each intermediate molecule as it arises during search?*

Answering this research question is non-trivial. A seemingly straightforward solution is to learn from abundant annotations that explicitly characterize the compatibility between molecules and SSR models; however, such fine-grained supervision is inaccessible. Alternatively, the planner could infer model-molecule compatibility autonomously during search, yet rewards that are observed only upon completion of an entire multi-step retrosynthetic pathway are sparse and delayed. Under such weak supervision, attributing credit to individual transition choices and isolating the contribution of a specific SSR model at each step remains challenging.

Recent advances have demonstrated that LLM agents can learn strategic tool-calling capability solely through environmental interaction, even in the absence of ground truth supervision (Zeng et al., 2025a; Qian et al., 2025). Parallel research has established both empirical (Sun et al., 2024; Kim et al., 2025; Zhang et al., 2025) and theoretical (Piotrowski et al., 2025) foundations for characterizing LLM agents as POMDP solvers where the model's context serves as a proxy for the belief state. Inspired by these developments, we propose an LLM-empowered multi-step retrosynthesis agent named *RetrOrchestrator*, which first formulates multi-step retrosynthesis as a POMDP where the agent must dynamically select the optimal SSR model as a tool along with the molecule to expand. To overcome

the aforementioned sparse reward problem, we introduce *Scaffold-aware Group-Relative Policy Optimization (SA-GRPO)*. SA-GRPO trains the agent to infer different SSR models' skill disparity based on their step-wise advantage when applied to scaffold-similar molecule states.

In summary, our key contributions are three-fold.

- *(LLM-empowered retrosynthesis planning agent)* We propose the first LLM-based retrosynthesis planning agent that enables a POMDP formulation of multi-step retrosynthesis, where the LLM's evolving context serves as an implicit belief state. This formulation allows the planner to learn from historical search feedback to dynamically orchestrate a set of SSR models.
- *(Curriculum-guided and scaffold-aware reinforcement learning)* To make LLM-based planning effective under sparse and delayed rewards, we propose SA-GRPO, which leverages scaffold-level clustering to assign credit to different SSR models at each transition. To bridge the gap between natural language reasoning and retrosynthetic decision-making, we further train SA-GRPO with a curriculum that progressively transitions the agent from simple chemical rule adherence on easy molecules to strategic planning on hard molecules.
- *(Pareto-optimal performance)* On the Retro*-190 benchmark, RetrOrchestrator establishes a new state-of-the-art performance, consistently outperforming static SSRs and their ensembles under equal computation budgets. Besides, it achieves a new Pareto frontier in terms of success rate versus model-query cost, demonstrating its effectiveness as a practical multi-step planning agent.

## 2. Preliminaries

**Multi-step Retrosynthesis Planning MDP Formulation** As illustrated in Figure 2, the retrosynthetic planning process is conventionally formulated as an MDP defined by the tuple $(\mathcal{S}, \mathcal{A}, \mathcal{T}, \mathcal{R})$. A state $s_t \in \mathcal{S}$ represents the current configuration of the search tree $M_t$, and feasible actions $a_t \in \mathcal{A}$ correspond to selecting a molecule $m_t \in M_t$ for decomposition. Such decomposition is performed by the transition model, an SSR model $\mathcal{T}(s_t, a_t, s_{t+1})$ that, given action $a_t$, transforms state $s_t$ into $s_{t+1}$. The reward function $\mathcal{R}(s_t, a_t, s_{t+1})$ returns a scalar reward $r_t$ for the action taken at time step $t$.

**Group Relative Policy Optimization (GRPO)** Standard policy gradient methods aim to update the policy $\pi$ with parameters $\theta$ by maximizing the expected return, i.e.,

$$\mathcal{L}_{PG}(\theta) = \mathbb{E}_{\tau \sim \pi_\theta(\tau)}[\mathcal{R}(\tau)],$$

where $\tau$ denotes a trajectory generated by the policy $\pi_\theta$ and $\mathcal{R}(\tau)$ is its reward. However, such raw reward objective leads to high-variance and unstable gradient estimates.

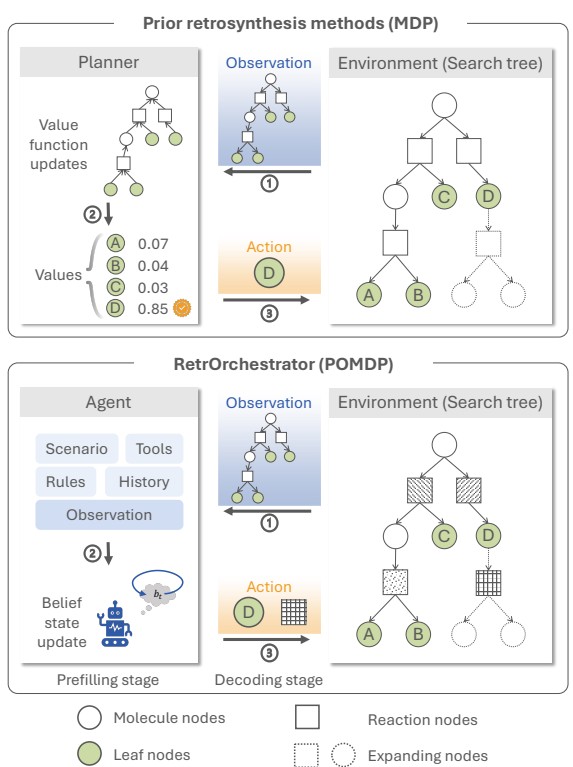

*Figure 2.* Comparison of our proposed methodology, formulated as POMDP, versus prior approaches, formulated as MDPs.

To address this issue, Proximal Policy Optimization (PPO) (Schulman et al., 2017) replaces raw rewards with advantage estimates $A$ and introduces a clipped surrogate objective:

$$\mathcal{L}_{\text{clip}}(\theta) = \mathbb{E}\left[\min\left(\text{is}(\theta)A, \text{clip}\left(\text{is}(\theta), 1-\epsilon, 1+\epsilon\right)A\right)\right], \quad (1)$$

where $\text{is}(\theta) = \frac{\pi_\theta(a|s)}{\pi_{\theta_{\text{old}}}(a|s)}$ is the importance sampling ratio, $\epsilon$ is the clipping coefficient. This stabilization, however, relies on an external critic to estimate advantages $A$, which introduces additional computational overhead.

Recent group-based advantage estimation methods, including REINFORCE Leave-One-Out (RLOO) (Kool et al., 2019), REINFORCE++ (Hu et al., 2025), and GRPO (Shao et al., 2024), have been proposed to eliminate such an external critic model by sampling a group of $G$ rollouts from each initial state $s_0$ under the current policy $\pi_\theta$, i.e., $\mathcal{G} = \{\tau_1, ..., \tau_G\} \sim \pi_\theta \mid s_0$. The advantage $A_i$ for each trajectory $\tau_i$ within the group is computed by normalizing its episodic reward $\mathcal{R}(\tau_i)$ relative to the group's performance:

$$A_i^E = \frac{\mathcal{R}(\tau_i) - \mu(\mathcal{R}(\tau_1), \mathcal{R}(\tau_2), \ldots, \mathcal{R}(\tau_G))}{\sigma(\mathcal{R}(\tau_1), \mathcal{R}(\tau_2), \ldots, \mathcal{R}(\tau_G))}, \quad (2)$$

where the superscript $E$ denotes trajectory-level (episodic) advantage, and $\mu, \sigma$ are the mean and standard deviation over the group.

## 3. Related Work

**Multi-step Retrosynthesis Planning** Multi-step retrosynthesis planning, pioneered by Corey (1967), has traditionally been tackled with heuristic search such as MCTS (Segler et al., 2018) and A*-like algorithms (Chen et al., 2020), with variants optimizing search via tailored value functions or experience-guided metrics (Yu et al., 2022; Hong et al., 2023; Yuan et al., 2024; Schreck et al., 2019). These approaches commit to a static SSR or, at best, a fixed ensemble (Maziarz et al., 2025a) or manual selection (Tu et al., 2025). More recently, LLM-based planners with Retrieval-Augmented Generation (RAG) (Chang et al., 2024) demonstrated the value of leveraging structurally similar routes as references, further refined by evolutionary search (Wang et al., 2025a) and A* over LLM-generated pathways (Song et al., 2025b), but rely on hundred-billion-parameter LLMs incurring high latency. Despite their differences in search strategy, all of these planners share a structural limitation: the transition operator is fixed before search and cannot adapt to the chemical demands of the molecules encountered along the route. Independent benchmarking (Maziarz et al., 2025b) further documents pronounced model heterogeneity across functional-group classes, suggesting that intermediate-conditioned model selection is a natural axis of improvement left untouched by prior planners.

**Tool-Calling Agents** Tool-calling agents trained by trajectory-level RL (Zeng et al., 2025b; Qian et al., 2025; Feng et al., 2026; Wang et al., 2025c; Feng et al., 2025; Li et al., 2025; Wang et al., 2025b; Song et al., 2025a) increasingly underpin complex tool use. Group-based variants like GRPO (Shao et al., 2024) improve efficiency but rely on coarse episodic rewards, often requiring complex reward engineering (Feng et al., 2026) or extensive SFT (Liu et al., 2025; Zeng et al., 2026) for agentic tasks. Recent step-level credit assignment via strict state grouping (Feng et al., 2025) or adaptive extra rollouts (Dong et al., 2026) improves long-horizon performance but, in retrosynthesis planning, the former degrades to vanilla GRPO under sparse state matches while the latter imposes prohibitive compute.

**Positioning of RetrOrchestrator** Two gaps emerge across the above lines of work. First, prior retrosynthesis planners commit to a fixed SSR and leave the pronounced *skill disparity* across molecule states unexploited, even though no single SSR is universally optimal (Figure 1). Second, generic tool-calling agents either rely on coarse trajectory-level rewards or impose state-matching criteria that are too sparse to support stable step-level credit in the vast chemical space. RetrOrchestrator closes both gaps simultaneously: it orchestrates SSRs as tools under a POMDP formulation and replaces exact state matching with chemistry-aware scaffold grouping.

# 4. Methodology

To address the limitation of static SSR choice in traditional MDP formulation, we propose the first POMDP formulation that enables dynamic model selection. We begin by introducing the conceptual framework, then detail its practical implementation with our design choice for the LLM agent, and finally present our training algorithm, Scaffold-Aware GRPO (SA-GRPO).

## 4.1. Our Conceptual Formulation: A POMDP

To account for skill disparity among SSR models, we conceptualize multi-step retrosynthesis planning as a POMDP with a fixed horizon $T$. This formulation extends the previous MDP to handle uncertainty in selecting transition models, where a **hidden belief state** ($b_t$) captures the agent's evolving understanding of SSR models' capabilities. It is formally defined by $(\mathcal{S}, \mathcal{A}, \mathcal{T}, \mathcal{R}, \Omega, \mathcal{O})$ with the following key modifications. Each state $s_t = M_t$ remains the synthesis planning tree ($M_t$) to date at time $t$, where we adopt the AND-OR tree following (Chen et al., 2020). The action space $\mathcal{A}$ is significantly expanded, where each $a_t \in \mathcal{A}$ includes the molecule selected, the SSR model used, and the number of reactions to generate. Unlike MDP-based planners such as Retro*, where stopping is handled by an external search algorithm, our policy is itself the planner and must decide when to halt. We therefore add TERMINATE to the action space to explicitly signal route completion. After each action, the agent receives its outcomes as an observation $o_t$, which is the set of expandable molecules in $M_t$ in retrosynthesis planning, from the observation space $\mathcal{O}$ via an observation function $\Omega : \mathcal{S} \rightarrow \mathcal{O}$. Under the above formulation, the goal of a POMDP planner is to generate a retrosynthesis route $\tau$. Each trajectory $\tau$ consists of a sequence of observation, action, and reward triplets, i.e., $\tau = \{(o_t, a_t, r_t), t \in [0, T]\}$. We measure the quality of a retrosynthesis route by its total discounted reward, $\mathcal{R}(\tau) = \sum_{t=0}^{T} \gamma^t r_t = \sum_{t=0}^{T} \gamma^t \mathcal{R}(s_t, a_t)$, where $\gamma$ is the discount factor that determines the importance of future rewards. Under this formulation, the agent leverages observations to continuously update its internal belief about model-molecule compatibility, thereby enabling an effective and dynamic SSR model selection strategy.

## 4.2. LLM Agent as a POMDP Planner

At timestep $t$, the POMDP planner first updates its belief state $b_t$ based on the current observation $o_t$ via function $f$, and then generates an action $a_t$ following a policy $\pi_\theta$ conditioned on $b_t$, i.e., $a_t = \pi_\theta(b_t)$ where $b_t = f(o_t)$. Note that such update-generation paradigm is highly analogous to the prefilling-decoding paradigm in LLM inference (detailed comparison is shown in Appendix A). In light of this, several studies (Kim et al., 2025; Zhang et al., 2025; Sun et al.,

2024) have proposed using LLMs' prefilled KV cache as the implicit belief state, and Piotrowski et al. (2025) has further theoretically validated this connection.

Motivated by this connection, we treat the LLM's prefilled KV cache as the implicit belief state $b_t$. For context tokens $X_{\text{ctx}}$ (prompt $p$ and observation $o_t$), $b_t$ is formed through the forward pass: $b_t = f_\theta(X_{\text{ctx}})$ and the output token sequence $Y_{\text{act}}$ is generated based on $b_t$ in sequence, specifying the molecule to expand, the single-step model to use, and the number of predictions to generate.

While LLM agents are implicit POMDP planners, they are primarily trained to encode broad, general-purpose knowledge. As a result, relying solely on their context window to infer molecule-model compatibility rules and chemical nuances often yields sub-optimal planning outcomes. Many aspects critical to retrosynthetic decision-making, including long-horizon planning dependencies and skill disparity among SSR models, are not explicitly represented in general LLM pretraining and cannot be reliably inferred from context alone at inference time. This gap necessitates explicit reinforcement learning, which we will introduce in the next subsection, to internalize retrosynthesis-relevant knowledge into the model parameters $\theta$. During inference, the parameters $\theta$ are held constant and the LLM agent updates its belief state simply as it encounters new observations.

## 4.3. Curriculum Training with Scaffold Awareness

Following recent tool-calling literature (Feng et al., 2026; Zeng et al., 2025a; Qian et al., 2025), we train the policy $\pi_\theta$ on a group of $G$ rollouts from each initial state $o_0$ under the current policy $\pi_\theta$ using the objective of

$$\mathcal{L}_{RL}(\theta) = \mathbb{E}_{\mathcal{T}} \left[ \mathcal{L}_{\text{clip}} + \beta_{\text{KL}} \mathcal{L}_{\text{KL}} \right], \tag{3}$$

$$\text{with } \mathcal{L}_{\text{KL}} = \frac{1}{GT} \sum_{g,t} \log \frac{\pi_\theta(a_t | o_t, b_t)}{\pi_{\theta_{\text{ref}}}(a_t | o_t, b_t)}. \tag{4}$$

Following Yu et al. (2025), we set different clipping coefficients $\epsilon$ for the lower and upper bound in Equation (1), denoted as $\epsilon_l$ and $\epsilon_h$, respectively. We assign a reward of $+1$ to the terminal action when a valid synthesis route is found; otherwise, any invalid action incurs a penalty of $-0.1$ and all other valid actions receive a neutral reward of $0$. An action is considered invalid if it cannot be executed by the planning environment, including malformed / unparsable output, selecting a non-expandable molecule, choosing an unavailable SSR or illegal prediction count, invoking an SSR that returns no expansion, or issuing TERMINATE before a valid route is found.

As for computation of the advantage $A$ in Equation (1), we propose both an episode-level advantage that evaluates at the trajectory level and a step-level advantage that assigns credit to fine-grained actions, as illustrated in the bottom

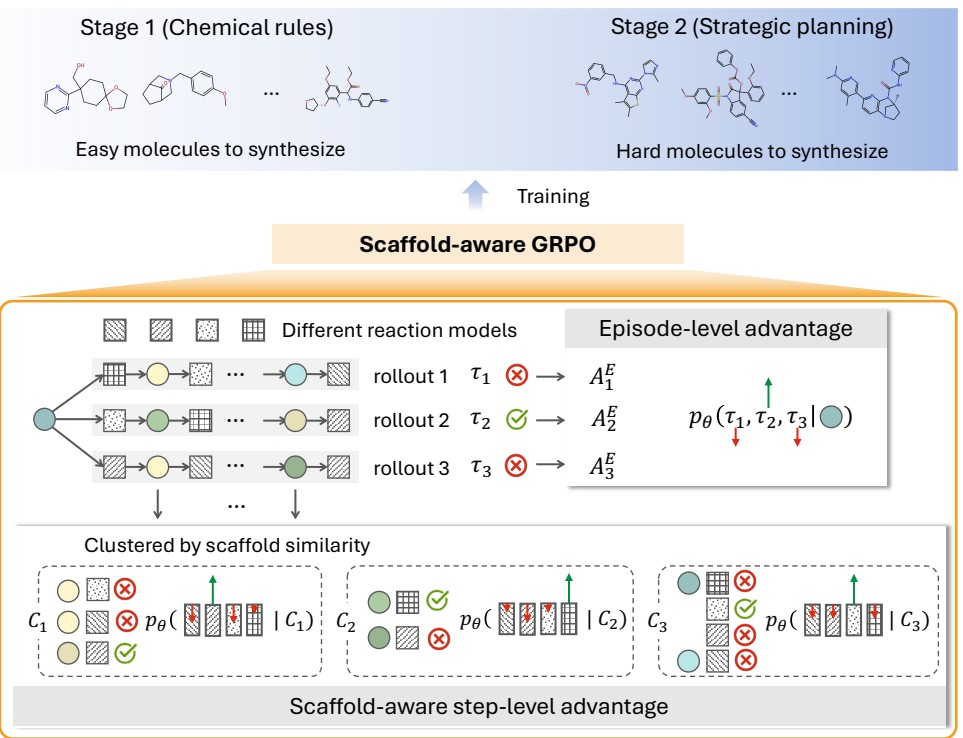

*Figure 3.* Overview of the RetrOrchestrator training algorithm and pipeline. The bottom panel illustrates the scaffold-aware GRPO mechanism, which leverages both episode-level and step-level advantages. Step-level advantage is computed over clusters of observations with molecules sharing similar scaffolds, explicitly steering the search toward more feasible solutions. The training follows a two-stage curriculum: Stage 1 emphasizes chemical rule adherence using simpler targets (SCScore < 4), while Stage 2 focuses on complex reasoning for more challenging molecules (SCScore > 4).

panel of Figure 3. Given the initial observation $o_0$, we have a rollout group $\mathcal{G}$ and corresponding episode-level advantages $A_i^E$ (see Equation (2)), which serve as the training signals to update the policy parameters $\theta$ and shift the probability distribution $p_\theta(\cdot \mid o_0)$ toward high-advantage trajectories (e.g., $\tau_2$ in Figure 3). However, this approach lacks a fine-grained signal to identify which specific actions, derived from different SSR models, are optimal at intermediate stages.

Inspired by the credit assignment proposed by Feng et al. (2025) and the proven success of RAG in multi-step retrosynthesis planning (Chang et al., 2024; Wang et al., 2025a; Song et al., 2025b), we propose Scaffold-aware GRPO (SA-GRPO), which introduces a step-level advantage $A^S$ by clustering intermediate observations based on the similarity of their molecular scaffold sets. Exact molecule matching in retrosynthesis creates sparse, high-variance credit signals because identical intermediates rarely recur across different paths. This creates a "sparsity" problem where 20% of observations form singleton clusters, leading to high-variance and unstable policy optimization. Moving to scaffold-based clustering solves this by grouping molecules by structural analogy rather than identity. By increasing group density and reducing singletons to under 6%, this approach allows

the model to exploit recurrent patterns across related scaffolds, resulting in a more stable and semantically meaningful credit signal. Concretely, the proposed clustering procedure consists of three steps.

**Scaffold Extraction.** Given the observation of the $i$-th trajectory at timestep $t$, $o_t^i$ comprising a set of molecules $M_t^i$, we extract the Bemis–Murcko scaffolds (Bemis & Murcko, 1996) for each molecule $m \in M_t^i$ using RDKit (Landrum & contributors, 2006). This operation isolates the core structural frameworks from their substituents, yielding a set of scaffolds $\mathcal{K}_t^i$:

$$\mathcal{K}_t^i = \{\phi(m) \mid m \in M_t^i\}, \tag{5}$$

where $\phi(\cdot)$ denotes the scaffold extraction function.

**Similarity Computation.** We quantify the pairwise similarity between scaffold sets across trajectories using the Tanimoto coefficient, which is equivalent to the Jaccard index when applied to molecular fingerprints. This choice is motivated by two considerations. First, scaffold-level similarity captures the synthetic invariants that drive the strategic disconnection (Bemis & Murcko, 1996), while remaining robust to peripheral functional-group changes that often correspond to interchangeable late-stage steps and therefore

*Table 1.* Performance comparison on Retro*-190 dataset of retrosynthesis planning MDP baselines, LLM planner baselines and RetrOrchestrator trained under various settings. For LLM baselines, we report the best results from their respective papers (GPT-4o for Wang et al. (2025a), Deepseek-R1 for Song et al. (2025b)) using an iteration budget of 100 to ensure a fair comparison. Full results with Retro*-0 baselines can be found in Table A.2.

| | Planner | Setting | Success Rate ↑ | Avg. Wallclock Time (s) | Avg. No. of Expansions | Avg. No. of Unique Routes | Avg. Length of Routes | Avg. Util. Ratio ↑ | Avg. SCScore↑ |
|---|---|---|---|---|---|---|---|---|---|
| MDP Baselines | Retro* | LocalRetro | 55.2632% | 1.8497 | 25.6190 | 1.6476 | 4.7061 | 17.4041% | 3.8003 |
| | | RetroKNN | 54.7368% | 2.3107 | 24.2981 | 1.6442 | 4.6858 | 16.4489% | 3.7254 |
| | | Chemformer | 50.0000% | 68.0708 | 25.6632 | 1.0316 | 6.2432 | 21.6271% | 3.9287 |
| | | Root Aligned | 38.9474% | 208.1969 | 5.5811 | 1.0811 | 3.4678 | 60.0392% | 3.8011 |
| | | MEGAN | 51.5789% | 2.6431 | 28.1224 | 1.0204 | 5.3616 | 17.6664% | 3.8740 |
| | | Graph2Edits | 57.8947% | 17.1373 | 19.7545 | 1.0091 | 6.7759 | 27.6022% | 3.7649 |
| | | EditRetro | 80.5263% | 163.1328 | 20.6732 | 3.5686 | 10.1126 | 30.0586% | 3.8623 |
| | | Ensemble | 77.8947% | 183.4332 | 4.4797 | 2.7230 | 4.6268 | 63.3300% | 3.8670 |
| | MCTS | LocalRetro | 55.2632% | 0.9934 | 17.7905 | 1.8190 | 5.4840 | 24.3442% | 3.8456 |
| | | RetroKNN | 62.6316% | 1.8258 | 19.7059 | 1.6218 | 7.5653 | 31.4685% | 3.8925 |
| | | Chemformer | 30.0000% | 48.6835 | 20.9649 | 1.0526 | 4.6535 | 24.3523% | 3.8567 |
| | | Root Aligned | 38.4211% | 160.3266 | 13.9315 | 1.0822 | 2.3333 | 16.8248% | 3.6168 |
| | | MEGAN | 37.3684% | 1.6202 | 20.9718 | 1.0282 | 5.7027 | 28.5135% | 3.7593 |
| | | Graph2Edits | 47.3684% | 16.8124 | 21.5111 | 1.0111 | 4.8056 | 22.5260% | 3.7415 |
| | | EditRetro | 68.9474% | 78.3119 | 21.8626 | 3.3740 | 5.1388 | 15.9961% | 3.7392 |
| | | Ensemble | 64.7368% | 216.9616 | 4.0488 | 2.1789 | 4.4465 | **89.2677%** | 3.8167 |
| POMDP | LLM Planners | Qwen2.5-0.5B | 0.5263% | 123.4800 | 42.0000 | 1.6667 | 1.0000 | 4.7619% | 3.2476 |
| | | Qwen2.5-1.5B | 9.4737% | 104.9112 | 15.5556 | 1.5294 | 2.8077 | 27.1429% | 3.2476 |
| | | Qwen2.5-3B | 63.6842% | 295.8222 | 12.8512 | 1.5372 | 9.5753 | 67.8457% | 3.9105 |
| | | Wang et al. (2025a) | 64.7368% | NA | NA | NA | NA | NA | NA |
| | | Song et al. (2025b) | 90.5263% | NA | NA | NA | NA | NA | NA |
| | non-LLM Router | Herustics | 82.6316% | 78.7745 | 22.2439 | 1.2166 | 7.0553 | 23.4125% | 3.8385 |
| | RetrOrchestrator | GRPO | 61.9565% | 391.1371 | 8.1053 | 2.3587 | 5.4561 | 67.3160% | 3.7920 |
| | | GiGPO | 74.2105% | 192.7996 | 14.1348 | 2.9737 | 9.3050 | 65.8304% | 3.9272 |
| | | Stage 1 Only | 65.4255% | 219.7301 | 13.0000 | 1.9574 | 8.5610 | 65.8537% | 3.9105 |
| | | Stage 2 Only | 72.2222% | 90.0011 | 9.2614 | 1.9200 | 8.5124 | 78.8360% | **3.9702** |
| | | Scaffold-aware GRPO | **94.2105%** | 118.9155 | 15.3473 | 1.7929 | 9.0075 | 76.4002% | **3.9702** |

should not affect SSR-selection credit. Second, the Tanimoto coefficient remains well-defined when scaffold sets differ in cardinality across rollouts—the common case in retrosynthesis, where intermediate trees grow at different rates—and is symmetric in its arguments, so observations from rollouts of different lengths contribute to the same cluster on equal footing. For two observations $o_t^i$ and $o_{t'}^j$ where $o_{t'}^j$ denotes a different observation from trajectory $\tau_j$ at step $t' \neq t$, we define the similarity score as

$$\text{Sim}(o_t^i, o_{t'}^j) = \frac{\left| \mathcal{K}_t^i \cap \mathcal{K}_{t'}^j \right|}{\left| \mathcal{K}_t^i \cup \mathcal{K}_{t'}^j \right|}. \qquad (6)$$

**Clustering.** We initialize each $(o_t^i, a_t^i, r_t^i)$ triplet as an individual cluster and iteratively merge the two clusters that maximize the average inter-cluster similarity, i.e., two clusters are merged if

$$\frac{1}{|C_a||C_b|} \sum_{o_i \in C_a} \sum_{o_j \in C_b} \text{Sim}(o_t^i, o_t^j) > \delta, \qquad (7)$$

and the process terminates when the average similarity between all remaining candidate pairs of clusters falls below

$\delta$. We compute the stepwise reward as

$$r_t^k = \sum_{j=t}^{T} \gamma^{j-t} r_j^k, \quad \text{where} \quad r_T^k = \mathcal{R}(\tau_k), \qquad (8)$$

where $\tau_k$ is the corresponding trajectory. Denoting the cluster of $(a_t^i, r_t^i)$ as $C(a_t^i, r_t^i)$, the corresponding stepwise advantage for action $a_t^i$ is then given by

$$A_{i,t}^S = \frac{r_t^i - \mu\left(\{r_{t'}^j \mid (a_{t'}^j, r_{t'}^j) \in C(a_t^i, o_t^i)\}\right)}{\sigma\left(\{r_{t'}^j \mid (a_{t'}^j, r_{t'}^j) \in C(a_t^i, o_t^i)\}\right)}. \qquad (9)$$

The final advantage used for Equation (1) is

$$A_{i,t} = A_i^E + A_{i,t}^S. \qquad (10)$$

During training of SA-GRPO, we adopt a two-stage curriculum: we first train the policy on easier molecules to learn chemical rules, and then on hard molecules to learn strategic planning. Molecule difficulty is graded by the Synthetic Complexity Score (SCScore) (Coley et al., 2018). The rationale for such two-stage curriculum is to decouple fundamental rule adherence from high-level planning to maximize the sample efficiency. We present the overall pipeline in Figure 3.

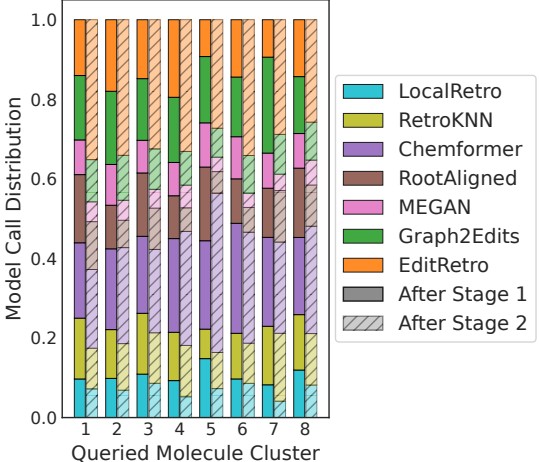

*Figure 4.* Evolution of RetrOrchestrator's model selection policy. Solid and hatched bars represent call distributions made by the model after Stage 1 and Stage 2 training, respectively.

## 5. Experiment

### 5.1. Setup

**Tools (Models) Available** We utilize a diverse set of SSR models as tools, including template-based (Local-Retro (Chen & Jung, 2021), RetroKNN (Xie et al., 2023)), template-free (Chemformer (Irwin et al., 2022), Root Aligned (Zhong et al., 2022)), and editing-based methods (MEGAN (Sacha et al., 2021), Graph2Edits (Zhong et al., 2023), EditRetro (Han et al., 2024)). More details about each SSR model can be found in Appendix N.

**Dataset** All SSR models were trained on the USPTO-50K dataset. We evaluated the performance on Retro* test dataset (Chen et al., 2020), containing 190 hard-to-synthesize molecules (Retro*-190). We also evaluated out-of-distribution (OOD) generalization on two PDB subsets, PDB-120 and PDB-600, containing 120 and 600 unseen hard-to-synthesize molecules respectively (Appendices B, F). We curated Stage 1 and Stage 2 training datasets using SCScore-based (Coley et al., 2018) stratified sampling on the Retro* training set. A total of 1,080 molecules with SCScore values lower than 4 were sampled for Stage 1 training, and 480 molecules with SCScore values higher than 4 were selected for Stage 2 training. The initial observations of these molecules were then collected as the training data. We utilized the same inventory of 23.1M purchasable molecules following Retro* (Chen et al., 2020).

**Baselines** We compared RetrOrchestrator against traditional MDP baselines, comprising combinations of various planning algorithms equipped with different portfolios of SSR models, and LLM planners, including out-of-the-box Qwen-2.5 models (0.5B, 1.5B, 3B) and specialized agents from Wang et al. (2025a); Song et al. (2025b). The planning algorithms evaluated were Retro* (Chen et al., 2020) and

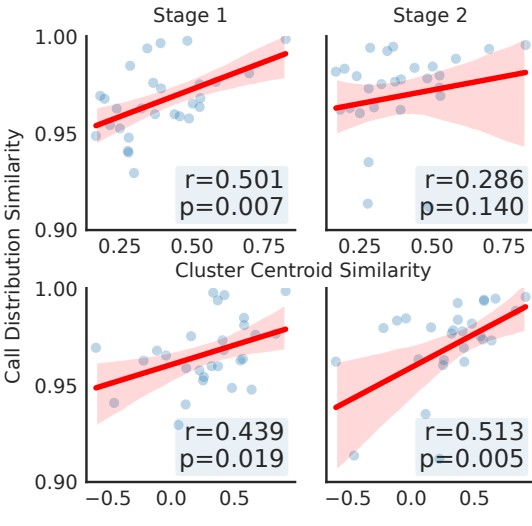

*Figure 5.* Correlation analysis of call distribution across clusters. (Top row) The correlation between cluster centroid similarity and call distribution similarity in Stage 1 and Stage 2. (Bottom row) The correlation between per-cluster performance similarity and call distribution similarity. The performance similarity is defined as the cosine similarity between Z-score normalized rankings of candidate models within each cluster.

an MCTS (Segler et al., 2018) variant from Maziarz et al. (2025b). We also evaluated Retro*-0, which is a variant of Retro* without a value function, and report the full results in Appendix C. For the MCTS baseline, we adopted the tuned parameters from Maziarz et al. (2025b) to account for its known sensitivity to its configuration (details in Appendix O.2). To ensure fairness, all MDP baselines were equipped with the same set of SSR models, which could be used either individually or as an ensemble (only for traditional planners). Following (Maziarz et al., 2025b), we applied the following limits per planning attempt: a **time limit** of 10 minutes, a maximum of 100 **expansions**, and up to 10 **reaction predictions returned per model per call**. An expansion was defined as a single call to a stand-alone model or one collective call to all members of an ensemble.

**Evaluation Metrics** We evaluated performance using the following metrics. For each setting, we performed 10 attempts with different random seeds. We report pass@10 as the success rate and report the average over all successful attempts for the other metrics. *1) Success Rate*: The percentage of target molecules with at least one valid synthesis route found across all attempts to measure *overall performance*. *2) Avg. Wall-clock Time*: The mean end-to-end time (in seconds) to find the first valid route, including all inference and planning overhead, serving as a measure of *efficiency*. *3) Avg. No. of Expansions*: The mean number of node expansions to find the first valid route, providing a measure of *hardware-independent efficiency*. *4) Avg. No. of Unique Routes*: The mean number of non-overlapping

routes found per molecule as in (Maziarz et al., 2025b), assessing *solution diversity*. *5) Avg. Length of Routes*: The mean number of reactions across all discovered routes, indicating *solution conciseness*. *6) Avg. Utilization Ratio*: The percentage of total expansions that belong to the final successful route, measuring *search efficiency*. *7) Avg. SCScore*: The mean SCScore (Coley et al., 2018) of final product molecules with routes found, evaluating *the capability to solve chemically complex targets*. The SCScore ranges from 1 (easily synthesizable) to 5 (hard to synthesize).

**Configuration** We trained RetrOrchestrator on top of the Qwen2.5-1.5B-Instruct model. Detailed hardware, software specifications and hyperparameters are available in Appendix O.

## 6. Results and Analysis

This section is structured to address three primary research questions (RQs):

**(RQ1) Does our method outperform existing planning and LLM-based baselines?**

**State-of-the-Art Success Rates.** As demonstrated in Table 1, our proposed method with 1.5B parameters achieves a state-of-the-art success rate of 94.21 % (95% bootstrap CI [90.53%, 97.26%]). This represents a significant improvement of 3.7 absolute percentage points compared to the strongest baseline proposed by Song et al. (2025b), backed by Deepseek-R1 with 671B parameters. RetrOrchestrator also outperforms a strong non-LLM state-dependent SSR router trained on the same molecule/search-graph features (82.63 %; McNemar $p = 3 \times 10^{-4}$), confirming that the gain stems from the joint policy over molecule, SSR, and prediction count rather than routing alone. Under a matched expansion budget, RetrOrchestrator further dominates the learned-value-function planner PDVN (Liu et al., 2023): **94.21% vs. 47.89%** at budget $= 10$, and **96.84% vs. 93.68%** at budget $= 50$ (Appendix G). Furthermore, 92.49 % of successful routes invoke at least two distinct SSRs, while only 7.51 % rely on a single SSR (Appendix H), showing that the learned policy does not collapse to one strong specialist. The OOD trend persists at a larger scale: on a 600-molecule PDB subset (PDB-600), RetrOrchestrator reaches 68.83 %, exceeding the best static baseline (Retro*+EditRetro, 59.67 %) by 9.16 absolute points and Retro*+Ensemble (47.33 %) by over 20; see Appendix B.

**Route Quality and Utilization.** Our method's 76.40 % utilization ratio is the second-highest overall (just shy of 89.27 % by MCTS+Ensemble), with the highest average SCScore (3.97) among solved molecules. This indicates that RetrOrchestrator does not just solve easy molecules, but tackles hard molecules with high efficiency.

**Stochastic Scaling.** The pass@k metric (Chen, 2021) across attempt budgets per molecule, shown in Figure A.6, indicates that RetrOrchestrator overtakes the top MDP baseline at $k = 2$ and remains ahead thereafter, confirming its search efficiency rather than reliance on large-$k$ randomness.

**(RQ2) Which components contribute to the performance gains and how does the agent's behavior evolve?**

**Training Stage Ablation.** Our analysis confirms that the full RetrOrchestrator paradigm is essential for increasing the 1.5B model's success rate from 9.5% to 94.2%. Stage 1 enforces chemical rule adherence, increasing action validity from 12.1% to 61.9% (matching the out-of-the-box 3B model) but does not yield strategic planning. In Stage 2, GiGPO (74.2%) outperforms vanilla GRPO (62.0%) by introducing discounted rewards, and SA-GRPO provides the definitive leap to 94.2% via fine-grained step-level advantages from scaffold-aware clustering. Per-component ablations on grouping, threshold, curriculum, and advantage weighting are reported in Appendix D–E.

**Emergent Model Selection Logic.** Analysis of Figure 4 reveals that the agent learns a systematic convergence toward robust models, most notably EditRetro. Across nearly all clusters, the transition from Stage 1 to Stage 2 involves a significant expansion of the EditRetro invocation fraction.

**Capability-Aligned Routing Logic.** To further investigate the underlying rationale of the learned policy, we analyze the correlation between the call distribution made by RetrOrchestrator and cluster properties with a Spearman test and visualize the results in Figure 5. We observe the **shift from feature-bias to performance alignment**. In Stage 1, the call distribution is primarily driven by molecular characteristics, showing a strong correlation with cluster centroid similarity ($\rho = 0.501$, $p = 0.007$). Such dependence weakens significantly in Stage 2 ($\rho = 0.286$, $p = 0.140$). We also observe the **emergence of relative strength awareness**, where the correlation between call distribution similarity and per-cluster performance similarity increases from $\rho = 0.439$ ($p = 0.019$) in Stage 1 to $\rho = 0.513$ ($p = 0.005$) in Stage 2.

**Cluster-Specific Adaptation.** Cluster-specific deviations from the general trend show that the policy does not simply exploit the best SSR. For instance, in Cluster 5 the agent favors Chemformer over EditRetro; the Appendix P case study shows RetrOrchestrator succeeding by switching to Graph2Edits and Chemformer in later steps where Retro*+EditRetro fails.

**Adaptive Branching Behavior.** Beyond *which* SSR to invoke, the policy also learns *how many* predictions to request (no_pred, Appendix I). no_pred is grounded in search dynamics and chemistry rather than SSR identity: it correlates with the step index (Spearman $\rho = 0.21$), Hall–Kier

| (a) Scaffold-similarity threshold $\delta$ | | | | |
|---|---|---|---|---|
| $\delta$ | 0.80 | 0.90 | **0.95** | 0.999 |
| Success Rate (%) | 71.58 | 90.63 | **94.21** | 73.68 |
| (b) Grouping granularity (at $\delta = 0.95$) | | | | |
| Level | | Molecule | Search-graph | **Scaffold** |
| Success Rate (%) | | 74.21 | 88.42 | **94.21** |

*Table 2.* SA-GRPO cluster-baseline sweeps on Retro*-190. (a) Scaffold-similarity threshold $\delta$: both overly loose and overly strict thresholds degrade the baseline. (b) Grouping granularity at $\delta$=0.95: scaffold-level dominates molecule- and search-graph-level. Both sweeps point to the same final configuration.

$\alpha$ (+0.30), and piperidine (+0.27), while the SSR-identity effect is small ($\eta^2 = 0.017$). The policy also contracts the branch on motifs where the SSRs are unreliable (TPSA, lactam, pyridine all $\rho < -0.3$), saving calls where extra candidates would not help.

**SA-GRPO Cluster-Baseline Sweeps.** The SA-GRPO cluster baseline has two design knobs: the scaffold-similarity threshold $\delta$ that controls cluster formation, and the grouping granularity. Table 2 sweeps both at the final-method scale. Loose $\delta$ mixes heterogeneous states ($\delta$=0.80: 71.58%) and overly strict $\delta$ collapses to singletons ($\delta$=0.999: 73.68%); $\delta$=0.95 is the empirical sweet spot (singleton rate ∼6%, mean cluster size 7.43). Likewise, scaffold-level grouping dominates molecule- and search-graph-level grouping at the same $\delta$, indicating the gain comes from a *matched-granularity* baseline rather than from clustering per se. Curriculum-cutoff (Table A.3) and advantage-weighting (Table A.4) ablations are reported in Appendix E.

**(RQ3) What is the performance-resource trade-off?**

**Call Efficiency.** Figure 6 situates RetrOrchestrator against existing baselines in the success-rate–cost plane. Following Maziarz et al. (2025b), we count each ensemble expansion as a joint call to all seven SSRs, so the x-axis reports total SSR calls rather than raw expansion counts, putting ensembled and single-model planners on a comparable cost axis. Under this fair accounting, RetrOrchestrator solves targets with about 10 SSR calls on average—roughly half of what Graph2Edits or Chemformer require—and remains on the Pareto front, while the static ensemble is no longer Pareto-optimal once its per-expansion overhead is exposed.

**Pareto-optimal Latency.** The LLM introduces latency (triangles in Figure 6), yet time-to-solution stays within the same order of magnitude as the Ensemble baseline while delivering ∼15% absolute improvement in success rate. A wall-clock decomposition (Appendix L) attributes only 19.3 % of inference time to LLM decoding, with SSR queries (69.2 %) dominating, so further latency gains would come from optimizing the SSR stack rather than the orchestrator.

**(RQ4) When does the agent fail, and is the learned routing interpretable?**

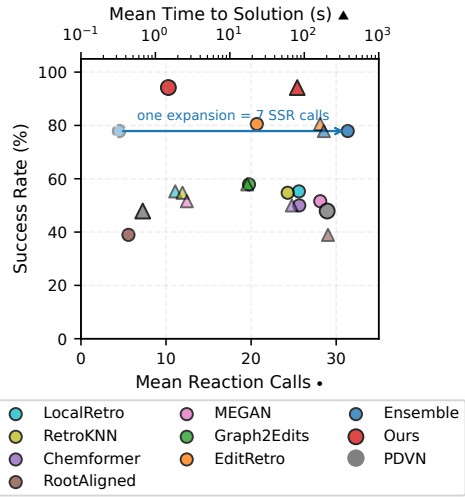

*Figure 6.* Efficiency–performance trade-offs between success rate and reaction-model calls to first solution (circles) versus time to first solution (triangles).

**When does the agent fail?** Of the 11 Retro*-190 failures, 5 are solved under a larger expansion budget and 10/11 lack any USPTO training neighbour at Tanimoto $\geq 0.7$; failed targets are also enriched in $sp^3$ centers (**0.60** vs. 0.46), stereocenters (**3.91** vs. 1.83), and rare motifs such as silyl and organotin (Appendix K). The profile thus reflects finite search budget and sparse SSR coverage in the OOD regime, not instability of the orchestration policy itself.

**Motif-conditioned routing.** We further probed whether the learned policy reduces to a static motif $\rightarrow$ SSR mapping (Table A.8, Appendix J): motif-conditioned call-share shifts are modest ($\Delta$ share $\leq 0.12$, mostly $\leq 0.05$), and for several subgroups the model with the largest shift is *not* the molecule-level dominant model. Routing is therefore driven by within-step search context, consistent with the POMDP formulation—orchestration value emerges from *dynamic* state-conditioned decisions, not a one-shot motif lookup.

## 7. Conclusion

We introduced RetrOrchestrator, an LLM-powered agent that reframes multi-step retrosynthesis as a POMDP, making the SSR choice part of the policy rather than a static design decision, motivated by the pronounced *skill disparity* across SSRs. We train this policy with SA-GRPO, which groups rollouts by Bemis–Murcko scaffold similarity for difficulty-matched advantage estimates and step-level shaping. RetrOrchestrator attains $94.21\,\%$ on Retro*-190 with $92.49\,\%$ of solved routes invoking two or more SSRs—neither a single-specialist collapse nor a static router—transfers to the OOD PDB-600 set, and lies on the Pareto front of wall-clock and SSR-call cost, evidence that *how* the transition space is navigated is itself a first-class lever.

## Acknowledgements

We thank Lang Feng for the insightful discussions during the rebuttal period.

## Impact Statement

This work introduces RetrOrchestrator, an RL framework that significantly enhances the success rate of multi-step retrosynthesis planning. By leveraging scaffold-aware advantage estimation (SA-GRPO), our method bridges the gap between algorithmic search and expert-like chemical intuition.

The primary positive impact is the potential acceleration of drug discovery and materials science, reducing the time and cost required to identify viable synthetic routes for novel functional molecules. Furthermore, the methodological innovation of using domain-specific structural similarity to densify sparse rewards offers a blueprint for applying RL to other complex, step-wise scientific optimization tasks.

While multi-step retrosynthesis planning algorithms could theoretically be misused for the synthesis of hazardous substances, we believe the benefits to the legitimate scientific community far outweigh these risks. We advocate for the integration of our framework with existing chemical safety protocols and regulatory screening databases to ensure responsible use in real-world laboratory settings.

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

## A. Analogy of POMDP Planner and LLM Agent

At timestep $t$, the POMDP planner first updates its belief state $b_t$ based on the current observation $o_t$ via function $f$, and then generates an action $a_t$ following a policy $\pi_\theta$ conditioned on $b_t$, i.e., $a_t = \pi_\theta(b_t)$ where $b_t = f(o_t)$. For the inference stage of an LLM agent, it first build a KV-cache via processing all the input tokens $X_{\text{ctx}}$, and then generate output tokens $Y_{\text{act}}$ based on the KV-cache. We can relate the prefilling-decoding paradigm of LLM inference as the standard process of a POMDP planner where the LLM agent builds an implicit belief state $b_t$ with the context tokens $X_{\text{ctx}}$. The context tokens $X_{\text{ctx}}$ include both an instructional prompt $p$ and the current observation $o_t$. In a decoder-only architecture, concretely, the implicit belief state is processed as a full forward pass on $X_{\text{ctx}}$ by the LLM with parameters $\theta$,

$$b_t = f_\theta(X_{\text{ctx}}) = \texttt{Decoder}_l(H_{l-1}; \theta_{l-1}) \text{ for } l = 1, ..., L,$$

where $\texttt{Decoder}_l$ is the $l$-th layer of a decoder block with parameter $\theta_{l-1}$, $H_l$ is the hidden states after processing by the $l$-th layer and the initial hidden states are processed with the LLM's embeding layer, i.e., $H_0 = \texttt{Embed}(X_{\text{ctx}})$. Conditioned on $b_t$, the agent then auto-regressively generates an action $a_t$, during which the policy $\pi_\theta$ defines the conditional probability of the action's token sequence $Y_{\text{act}}$:

$$\pi_\theta(a_t|b_t) = P(Y_{\text{act}}|b_t; \theta) = \prod_{j=1}^{L} P(y_j|b_t, y_1, \ldots, y_{j-1}; \theta).$$

## B. Out-of-Distribution Evaluation on PDBBind

We further evaluated the performance of previously trained RetrOrchestrator on a subset of 120 molecules randomly sampled from the Protein Data Bank dataset (PDB-120). This serves as an out-of-distribution (OOD) test to evaluate the generalization capability of our approach. Molecules in this subset display substantial functional-group complexity and may contain multiple competing reactive sites, suggesting they are more challenging to synthesize. As demonstrated in Table A.1, RetrOrchestrator shows robust generalization and leading performance on the PDB-120 out-of-distribution test set, achieving a superior success rate of 65.83% that significantly outperforms established baselines. Despite the inherent structural and functional-group complexity of PDB ligands, our method maintains high search efficiency, requiring only 8.52 average expansions to identify successful routes. Furthermore, the model achieves an utilization ratio of 49.18%, which outperforms all baselines except for the ensemble of SSRs, highlighting the efficiency of RetrOrchestrator.

*Table A.1.* Performance comparison of static planner Retro* and RetrOrchestrator on out-of-distribution PDB-120 dataset.

| Planner | Settting | Success Rate ↑ | Avg. Wallclock Time (s) | Avg. No. of Expansions | Avg. No. of Unique Routes | Avg. Length of Routes | Avg. Util. Ratio ↑ | Avg. SCScore↑ |
|---|---|---|---|---|---|---|---|---|
| Retro* | LocalRetro | 39.1667% | 18.1471 | 13.7021 | 1.3830 | 3.1339 | 24.0272% | 3.7347 |
| | RetroKNN | 43.3333% | 35.9412 | 21.3846 | 1.2885 | 3.9941 | 25.2336% | 3.7899 |
| | Chemformer | 37.5000% | 86.7788 | 10.2444 | 1.1778 | 3.6269 | 39.9671% | 3.7456 |
| | Root Aligned | 51.6667% | 112.6392 | 7.5237 | 1.5323 | 3.2381 | 40.6606% | 3.8391 |
| | MEGAN | 46.6667% | 12.2548 | 11.5893 | 1.3750 | 4.2138 | 28.1158% | 3.8074 |
| | Graph2Edits | 53.3333% | 22.7993 | 17.5469 | 1.2656 | 6.4272 | 24.8277% | **3.8506** |
| | EditRetro | 59.1667% | 90.9604 | 22.3656 | 4.4225 | 8.1484 | 28.7812% | 3.7783 |
| | Ensemble | 45.8333% | 257.8624 | 2.3207 | 2.4364 | 3.0839 | **89.8245%** | 3.6794 |
| RetrOrchestrator | | **65.8333%** | 210.2934 | 8.5190 | 1.4237 | 4.9524 | 49.1828% | 3.6453 |

## C. Additional Experiments and Analysis on Retro*-190

We report the full evaluation results on Retro*-190 dataset including Retro*, Retro*-0, MCTS baselines, LLM planners as well as ablated versions of RetrOrchestrator in Table A.2.

## D. Sensitivity Analysis of $\delta$ and Grouping Strategy

We cluster all observations in the rollout buffer using a similarity threshold $\delta$ to build a cluster-wise baseline for step-level advantages, aiming to reduce estimator variance. A lower $\delta$ yields overly large clusters that mix heterogeneous states,

*Table A.2.* Full performance comparison of retrosynthesis planning MDP baselines, LLM planner baselines and RetrOrchestrator trained under various settings. For LLM baselines, we report the best results from their respective papers (GPT-4o for Wang et al. (2025a), Deepseek-R1 for Song et al. (2025b)) using an iteration budget of 100 to ensure a fair comparison.

| | Planner | Settting | Success Rate ↑ | Avg. Wallclock Time (s) | Avg. No. of Expansions | Avg. No. of Unique Routes | Avg. Length of Routes | Avg. Util. Ratio ↑ | Avg. SCScore ↑ |
|---|---|---|---|---|---|---|---|---|---|
| MDP Baselines | Retro* | LocalRetro | 55.2632% | 1.8497 | 25.6190 | 1.6476 | 4.7061 | 17.4041% | 3.8003 |
| | | RetroKNN | 54.7368% | 2.3107 | 24.2981 | 1.6442 | 4.6858 | 16.4489% | 3.7254 |
| | | Chemformer | 50.0000% | 68.0708 | 25.6632 | 1.0316 | 6.2432 | 21.6271% | 3.9287 |
| | | Root Aligned | 38.9474% | 208.1969 | 5.5811 | 1.0811 | 3.4678 | 60.0392% | 3.8011 |
| | | MEGAN | 51.5789% | 2.6431 | 28.1224 | 1.0204 | 5.3616 | 17.6664% | 3.8740 |
| | | Graph2Edits | 57.8947% | 17.1373 | 19.7545 | 1.0091 | 6.7759 | 27.6022% | 3.7649 |
| | | EditRetro | 80.5263% | 163.1328 | 20.6732 | 3.5686 | 10.1126 | 30.0586% | 3.8623 |
| | | Ensemble | 77.8947% | 183.4332 | 4.4797 | 2.7230 | 4.6268 | 63.3300% | 3.8670 |
| | Retro*-0 | LocalRetro | 46.3158% | 2.6977 | 23.1250 | 1.7273 | 4.2163 | 16.5455% | 3.7165 |
| | | RetroKNN | 47.8947% | 4.6750 | 26.3187 | 1.6264 | 4.6283 | 18.3411% | 3.7214 |
| | | Chemformer | 46.8421% | 104.8982 | 25.9326 | 1.0337 | 6.1162 | 17.0491% | 3.8755 |
| | | Root Aligned | 48.9474% | 164.3959 | 13.7849 | 1.0645 | 4.2698 | 23.4693% | 3.7309 |
| | | MEGAN | 45.7895% | 4.3033 | 25.7701 | 1.0230 | 7.5885 | 16.1561% | 3.8313 |
| | | Graph2Edits | 57.3684% | 19.0907 | 19.5046 | 1.0092 | 7.1701 | 28.8427% | 3.7875 |
| | | EditRetro | 78.9473% | 83.5565 | 34.6301 | 3.6600 | 10.9767 | 21.6093% | 3.8579 |
| | | Ensemble | 62.1053% | 262.0556 | 3.9746 | 2.3475 | 4.0316 | 71.6452% | 3.7730 |
| | MCTS | LocalRetro | 55.2632% | 0.9934 | 17.7905 | 1.8190 | 5.4840 | 24.3442% | 3.8456 |
| | | RetroKNN | 62.6316% | 1.8258 | 19.7059 | 1.6218 | 7.5653 | 31.4685% | 3.8925 |
| | | Chemformer | 30.0000% | 48.6835 | 20.9649 | 1.0526 | 4.6535 | 24.3523% | 3.8567 |
| | | Root Aligned | 38.4211% | 160.3266 | 13.9315 | 1.0822 | 2.3333 | 16.8248% | 3.6168 |
| | | MEGAN | 37.3684% | 1.6202 | 20.9718 | 1.0282 | 5.7027 | 28.5135% | 3.7593 |
| | | Graph2Edits | 47.3684% | 16.8124 | 21.5111 | 1.0111 | 4.8056 | 22.5260% | 3.7415 |
| | | EditRetro | 68.9474% | 78.3119 | 21.8626 | 3.3740 | 5.1388 | 15.9961% | 3.7392 |
| | | Ensemble | 64.7368% | 216.9616 | 4.0488 | 2.1789 | 4.4465 | **89.2677%** | 3.8167 |
| LLM Planners | | Qwen2.5-0.5B | 0.5263% | 123.4800 | 42.0000 | 1.6667 | 1.0000 | 4.7619% | 3.2476 |
| | | Qwen2.5-1.5B | 9.4737% | 104.9112 | 15.5556 | 1.5294 | 2.8077 | 27.1429% | 3.2476 |
| | | Qwen2.5-3B | 63.6842% | 295.8222 | 12.8512 | 1.5372 | 9.5753 | 67.8457% | 3.9105 |
| | | Wang et al. (2025a) | 64.7368% | NA | NA | NA | NA | NA | NA |
| | | Song et al. (2025b) | 90.5263% | NA | NA | NA | NA | NA | NA |
| RetrOrchestrator | | GRPO | 61.9565% | 391.1371 | 8.1053 | 2.3587 | 5.4561 | 67.3160% | 3.7920 |
| | | GiGPO | 74.2105% | 192.7996 | 14.1348 | 2.9737 | 9.3050 | 65.8304% | 3.9272 |
| | | Stage 1 Only | 65.4255% | 219.7301 | 13.0000 | 1.9574 | 8.5610 | 65.8537% | 3.9105 |
| | | Stage 2 Only | 72.2222% | 90.0011 | 9.2614 | 1.9200 | 8.5124 | 78.8360% | **3.9702** |
| | | Scaffold-aware GRPO | **94.2105%** | 118.9155 | 15.3473 | 1.7929 | 9.0075 | 76.4002% | **3.9702** |

increasing baseline bias; a higher $\delta$ produces many singletons, degenerating the cluster statistics and increasing variance. We set $\delta = 0.95$ which results in a singleton rate of 6% and a mean cluster size of 7.43. This indicates that most steps still have sufficient neighbours for a stable cluster baseline, while clusters remain small enough to improve within-cluster homogeneity—providing a favourable bias–variance trade-off.

The threshold sweep over $\delta \in \{0.8, 0.9, 0.95, 0.999\}$ and the comparison of molecule-, search-graph-, and scaffold-level grouping (at $\delta = 0.95$) are reported in the main text (Table 2). Loose thresholds mix heterogeneous states and produce a biased baseline; the strictest setting yields too many singletons and collapses to episodic-only estimation. Molecule-level grouping is too specific (only 80% of steps have any neighbour); search-graph-level grouping is competitive but lower; scaffold-level grouping is the most informative.

## E. Curriculum and Advantage-Weighting Ablations

**Two-stage curriculum.** Under the same 300-step budget, removing either curriculum stage causes a large drop (Table A.3). Stage 1 alone teaches chemical-rule adherence but lacks strategic planning; Stage 2 alone optimizes strategic planning without a useful KL reference. The full curriculum is critical.

**SCScore cutoff.** The cutoff splitting Stage 1 (easy) and Stage 2 (hard) is itself sensitive: 3.5 makes Stage 1 too weak to anchor Stage 2, while 4.5 leaves Stage 2 with too few hard molecules. The choice of 4.0 reflects the bias–variance trade-off

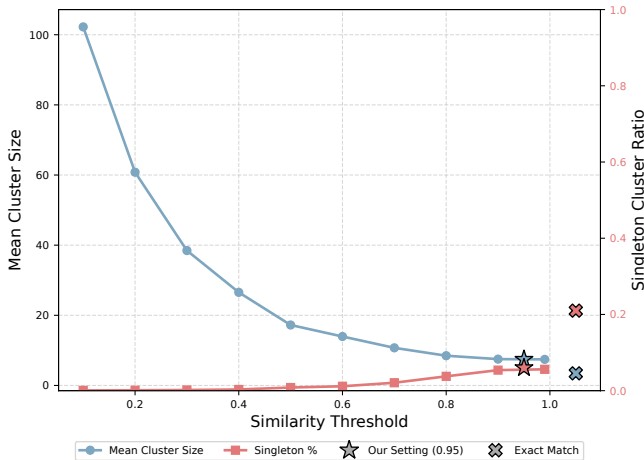

*Figure A.1.* Analysis of clustering metrics across varying similarity thresholds. The **mean cluster size** (blue) demonstrates a power-law decay, stabilizing as the threshold approaches 0.95. While the **singleton ratio** (red) increases with stricter thresholds, our setting maintains a low percentage ($\sim 6\%$) compared to the **exact match** baseline ($\sim 21\%$), preserving the relational density necessary for effective representation learning.

implied by the SCScore distribution.

**Episodic vs. step-level advantages.** SA-GRPO combines a route-level advantage $A_i^E$ with a scaffold-grouped step-level advantage $A_{i,t}^S$. Table A.4 shows that either component alone is insufficient, and over-weighting the step term collapses the policy. The balance—not the existence of a step-level term—drives the gain.

| Setting | Success Rate (%) |
|---|---|
| Stage 1 only | 65.43 |
| Stage 2 only | 72.22 |
| Full curriculum (cutoff 3.5) | 74.26 |
| Full curriculum (cutoff 4.5) | 72.11 |
| **Full curriculum (cutoff 4.0)** | **94.21** |

*Table A.3.* Curriculum ablation under a 300-step budget. Both stages and a cutoff near 4.0 are necessary.

| Advantage formulation | Success Rate (%) |
|---|---|
| Episodic only ($A_i^E$) | 74.21 |
| Step-level only ($A_{i,t}^S$) | 72.22 |
| Over-weighted step ($A_i^E + 5A_{i,t}^S$) | 60.00 |
| **Equal weights ($A_i^E + A_{i,t}^S$)** | **94.21** |

*Table A.4.* Ablation of advantage weighting in SA-GRPO. Both terms are required; balance matters.

# F. Larger-Scale PDB Evaluation

To check that the OOD trend in Table A.1 is not an artefact of the 120-molecule subset size, we evaluate on a 600-molecule PDB subset (PDB-600) sampled with the same protocol. RetrOrchestrator reaches $68.83\%$, consistent with the PDB-120 result ($65.83\%$) and well above the static baselines (Table A.5). The gap to static planners is preserved at $5\times$ the original benchmark scale. The absolute drop relative to Retro\*-190 is consistent with prior findings on retrosynthesis OOD generalization (Yu et al., 2024): PDB ligands depart substantially from the USPTO training distribution of every SSR, so the residual failures reflect OOD chemistry and limited SSR coverage rather than instability of the orchestration policy.

# G. Budget-Matched Comparison with PDVN

We compare RetrOrchestrator against PDVN (Liu et al., 2023), a strong learned-value-function planner for retrosynthesis, under matched expansion budgets on Retro\*-190 (Table A.6). At a tight budget of 10 expansions, RetrOrchestrator already

*Table A.5.* Performance comparison of static planner Retro* and RetrOrchestrator on the larger, out-of-distribution PDB-600 dataset.

| Planner | Setting | Success Rate ↑ | Avg. Wallclock Time (s) | Avg. No. of Expansions | Avg. No. of Unique Routes | Avg. Length of Routes | Avg. Util. Ratio ↑ | Avg. SCScore↑ |
|---|---|---|---|---|---|---|---|---|
| Retro* | LocalRetro | 43.6667% | 37.5267 | 15.3605 | 1.2060 | 4.0938 | 27.3817% | 3.2392 |
| | RetroKNN | 46.3333% | 29.7655 | 18.8381 | 1.2429 | 4.2463 | 22.1898% | 3.3317 |
| | Chemformer | 42.5000% | 69.3972 | 13.1136 | 1.2051 | 4.8476 | 31.4903% | 3.2373 |
| | Root Aligned | 53.3333% | 129.0730 | 4.7289 | 1.5845 | 3.0456 | **52.9986%** | 3.7929 |
| | MEGAN | 53.8333% | 37.6256 | 17.3868 | 1.2822 | 6.1130 | 28.3225% | 3.8533 |
| | Graph2Edits | 57.6667% | 31.4151 | 16.7231 | 1.2378 | 8.1000 | 28.6212% | **3.8978** |
| | EditRetro | 59.6667% | 9.9723 | 21.8405 | 1.0859 | 8.0631 | 30.8621% | 3.5473 |
| | Ensemble | 47.3333% | 105.1823 | 4.4902 | 2.1013 | 3.7571 | **58.1338%** | 3.7847 |
| RetrOrchestrator | | **68.8333%** | 210.2934 | 8.5190 | 1.4237 | 4.9524 | 49.1828% | 3.6453 |

reaches its full success rate (94.21%), while PDVN is only at 47.89%—a ~2× gap. At a more generous budget of 50, the gap narrows but RetrOrchestrator still leads (96.84% vs. 93.68%). The comparison isolates the contribution of dynamic SSR orchestration from improvements purely due to a richer value function: even when both methods can search five times as far, dynamic tool routing remains beneficial.

| Expansion Budget | Method | Success Rate ↑ |
|---|---|---|
| 10 | PDVN | 47.89% |
| | **Ours** | **94.21%** |
| 50 | PDVN | 93.68% |
| | **Ours** | **96.84%** |

*Table A.6.* Budget-matched comparison between RetrOrchestrator and PDVN on Retro*-190. RetrOrchestrator dominates at the tight 10-expansion budget and remains ahead at 50 expansions.

## H. Multi-SSR Usage in Successful Routes

A natural concern is that the policy might collapse onto a single best SSR (e.g., EditRetro). We instead find that $92.49\%$ of successful routes invoke at least two distinct SSRs, while only $7.51\%$ rely on a single SSR (Figure A.2). Among multi-SSR routes, the most frequent late-stage switches (EditRetro $\rightarrow$ Graph2Edits or EditRetro $\rightarrow$ Chemformer) account for the head-to-head wins illustrated in Figure A.7 and Figure A.8.

## I. Adaptive Branching Factor

Beyond SSR selection, the policy also learns an adaptive number of predictions per call (`no_pred`). The branching factor increases with search depth (Spearman $\rho = 0.21$, $p < 10^{-3}$), so the agent requests broader expansion on later, harder surviving states. It also correlates with chemical descriptors and motifs (Table A.7): more candidates are requested when the target has high Hall-Kier $\alpha$ or piperidine substructures, and fewer when TPSA is high or lactam/pyridine motifs dominate. The direct association with queried SSR identity is weak ($\eta^2 = 0.017$), so the branching policy is chemically grounded rather than a proxy for SSR choice.

| Feature | Spearman $\rho$ |
|---|---|
| Search-step index | $+0.21$ |
| Hall-Kier $\alpha$ | $+0.30$ |
| Piperidine count | $+0.27$ |
| TPSA | $-0.35$ |
| Lactam count | $-0.40$ |
| Pyridine count | $-0.32$ |

*Table A.7.* Correlations between the predicted branching factor `no_pred` and search-stage / chemical descriptors on Retro*-190.

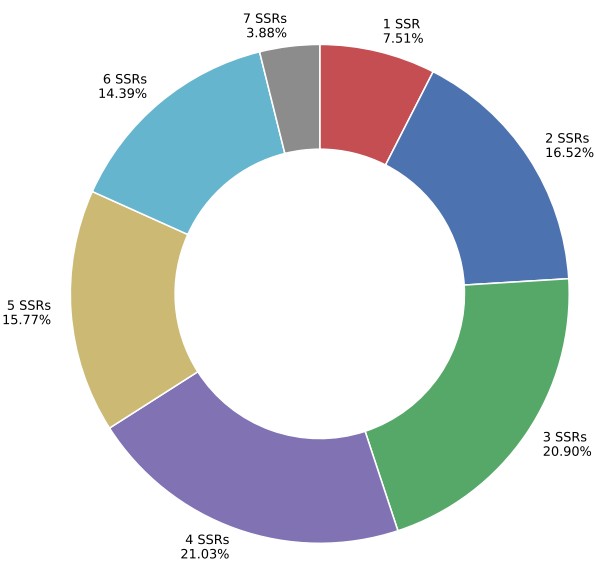

*Figure A.2.* Distribution of distinct SSRs invoked per successful route on Retro*-190. Only $7.51\%$ of solved routes use a single SSR; the bulk invoke two or more, supporting that the learned policy is not a single-specialist collapse.

## J. Motif-Conditioned SSR Preferences

Table A.8 reports motif-conditioned SSR preferences computed from per-step action posteriors on Retro*-190. For each motif/context we list the preferred SSR (the one whose call share increases most when the motif is present), the increase in call share $\Delta$, the number of molecules in the subgroup, and the fraction of molecules whose molecule-level dominant SSR matches the preference. The shifts are modest in absolute terms ($\Delta$ share at most $0.118$, mostly $\leq 0.05$), and for two subgroups the model with the largest call-share increase is *not* the molecule-level dominant model. This means the policy does not collapse to a static motif $\rightarrow$ SSR lookup: per-step routing is driven primarily by within-trajectory search context.

| Motif / context | Pref. SSR | $\Delta$ share | # mol | Dom. frac. |
|---|---|---|---|---|
| Piperidine | EditRetro | $+0.118$ | 31 | 0.581 |
| Chiral centers $> 0$ | EditRetro | $+0.039$ | 105 | 0.457 |
| Amide $> 0$ | EditRetro | $+0.031$ | 47 | 0.574 |
| Heteroatom-rich | Chemformer | $+0.086$ | 122 | 0.393 |
| Benzene-rich | Chemformer | $+0.047$ | 88 | 0.443 |
| High TPSA | Chemformer | $+0.045$ | 119 | 0.471 |
| Pyridine | Chemformer | $+0.043$ | 35 | $0.657^{\dagger}$ |
| Fused-ring topology | LocalRetro | $+0.009$ | 61 | $0.443^{\dagger}$ |

*Table A.8.* Motif-conditioned SSR preferences learned by RetrOrchestrator on Retro*-190. $\Delta$ share is the increase in the preferred SSR's call share conditional on the motif being present. Dom. frac. is the fraction of molecules in the subgroup whose molecule-level dominant SSR matches the preference. $^{\dagger}$ The molecule-level dominant SSR for these subgroups is EditRetro; the listed model is the one whose call-share *increase* is largest. The decoupling shows call allocation and dominant-model identity need not coincide.

## K. Failure-Mode Analysis

Among the 11 Retro*-190 failures, four mechanisms dominate. (i) **Search budget**: 5 of 11 are solved with a larger expansion budget; 1 remains unsolved by all methods. (ii) **Structural complexity**: failures have higher fraction sp3 (**0.60** vs. 0.46 for solved), more stereocenters (**3.91** vs. 1.83), and more aliphatic rings (1.91 vs. 1.56). (iii) **Rare motifs**: silyl (**18.2%** vs. 5.3%) and organotin (**9.1%** vs. 0.5%) are enriched in failures. (iv) **Weak SSR support**: maximum Tanimoto similarity to the USPTO training set is only 0.31–0.78 for failed targets, and 10 of 11 have no neighbour $\geq 0.7$. The failure profile thus reflects finite search budget and sparse SSR coverage in the OOD regime, not instability of the orchestration policy itself.

## L. Wallclock and Training-Cost Breakdown

**Inference.** Across Retro*-190, end-to-end inference time decomposes into **LLM decoding** (19.3 %), **SSR queries** (69.2 %), and **environment overhead** (11.5 %); see Figure A.3. Within SSR queries, Chemformer (43.3 %) and Root Aligned (27.2 %) dominate; the remaining five SSRs together account for under 30 %.

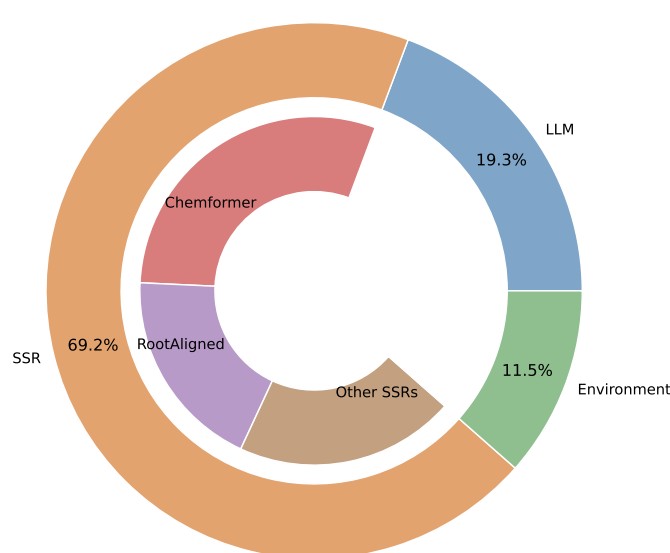

*Figure A.3.* Inference-time breakdown of RetrOrchestrator on Retro*-190. "Environment" covers maintaining the search graph, commercial-availability checks, and other overhead.

**Training.** One full RetrOrchestrator run takes roughly one week on 4×A100 GPUs. Rollout collection dominates wall time (62.4 %); scaffold clustering is negligible (0.1 %); all other components together account for the remainder (Figure A.4).

**Data leakage check.** Training molecules come exclusively from the Retro* training split and have zero exact-SMILES overlap with the Retro*-190 evaluation set. SCScore-stratified sampling preserves coverage across the easy/hard spectrum without leaking test targets (Figure A.5).

## M. Pass@k Analysis

We conducted an extended study on the number of parallel trials (k) in Figure A.6. The plot illustrates the scaling of success rate for RetrOrchestrator, who outperforms other baselins after $k = 2$, demonstrating high scalability.

## N. Details of Single-step Models Used

**Template-based** methods rely on a pre-extracted database of reaction templates, which define generalized reaction rules. The core challenge is to select the most appropriate template for a given product molecule. **LocalRetro** (Chen & Jung, 2021) enhances template selection by merging local and global molecular contexts with a graph neural network to capture the local chemical environment around potential reaction centers and a global attention mechanism that considers the influence of the entire molecule. **RetroKNN** (Xie et al., 2023) further augments it by applying a k-nearest neighbors (k-NN) algorithm on the extracted local context to improve performance on templates that are rare in the training dataset.

**Template-Free Methods** treat retrosynthesis as a sequence-to-sequence translation problem, typically converting a product's molecular representation into its reactants, to overcome the limitations of a fixed template library, **Chemformer** (Irwin et al., 2022) first adapts the powerful Transformer architecture, originally from natural language processing, to chemistry. By

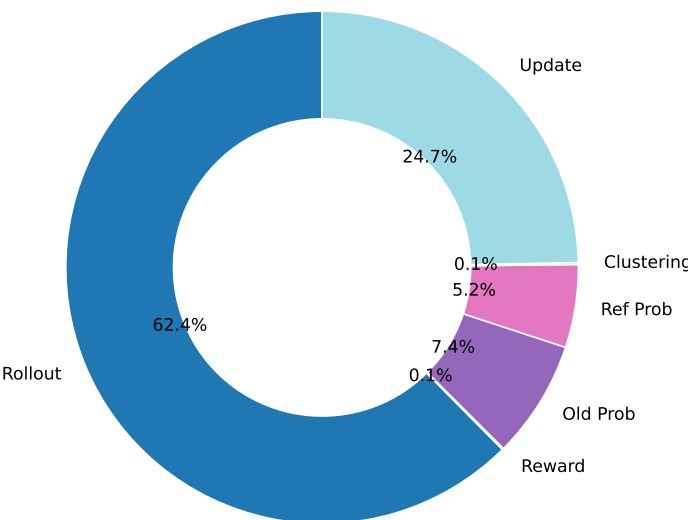

*Figure A.4.* Training-cost breakdown per optimization step. *Rollout* aggregates LLM inference, SSR calls, and environment overhead. *Clustering* covers scaffold extraction plus scaffold-based step-level advantage computation.

representing molecules as SMILES strings, it uses a neural machine translation method that "translates" a product sequence into the corresponding reactant sequences, learning the underlying rules of chemical transformations directly from data without explicit reaction templates. **Root Aligned** (Zhong et al., 2022) addresses a key challenge in SMILES-based models: representation variance. It introduces a method to align the SMILES strings of products and reactants so that corresponding atoms are in similar positions. This alignment simplifies the transformation task for the Transformer model, leading to improved prediction accuracy.

**Editing-Based Methods** reframes retrosynthesis as a process of modifying the product molecule to generate the reactants by predicting the specific changes required, such as breaking bonds or altering atoms. **MEGAN** (Sacha et al., 2021) conceptualizes reactions as a series of graph edits. It uses attention-based networks to predict a sequence of explicit actions that transform the product's molecular graph into the reactant graphs. **Graph2Edits** (Zhong et al., 2023) operates by first predicting the reaction center—the specific atoms and bonds to be edited within the product graph. Following this identification, it predicts the necessary edits and applies them to generate intermediates, which are subsequently completed into the final reactants. **EditRetro** (Han et al., 2024) offers a string-based alternative to graph editing. It directly modifies the product's SMILES string using an iterative sequence of edits. This process continues until the product's SMILES string is fully transformed into the SMILES strings of the reactants.

## O. Training Details

### O.1. Agent Training

The codebase for agent training is built upon verl-agent (Feng et al., 2025). Hyperparameters controlling the training is shown in Tab A.9. All experiments were conducted using NVIDIA A100 GPUs in a containerized CUDA 12.6 environment with data parallelism. Single-step model inference was supported by a dedicated single-GPU node. The software stack included Verl 0.3.1.dev, RDKit 2025.03, and vLLM 0.8.5, as a part of verl-agent 2025-06-12 update. For LLM-based approaches, including out-of-box LLMs and RetrOrchestrator, we set generation temperature to 0.6, top_p to 0.95, top_k to 20.

### O.2. Baseline Hyperparameters

We report the hyperparameter settings for MCTS-based baselines in Table A.10.

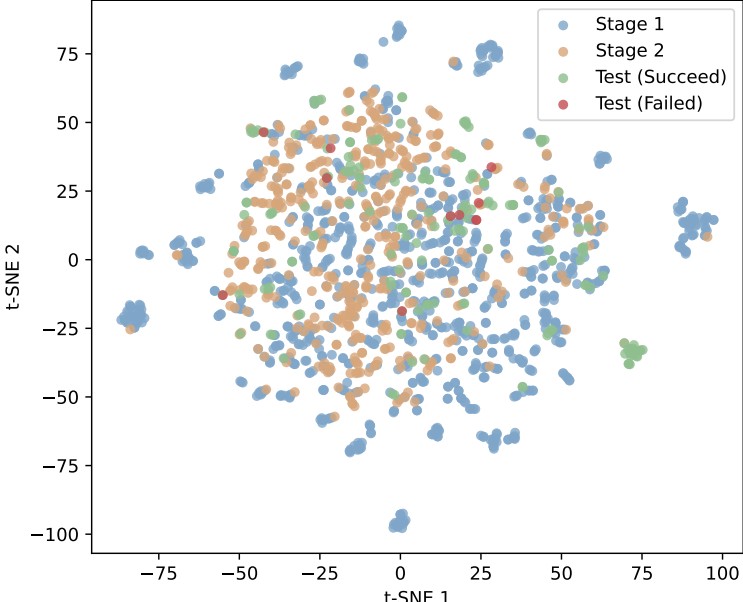

*Figure A.5.* Distribution of Stage 1 / Stage 2 training molecules and Retro*-190 test molecules over SCScore. The training pool covers both easy and hard regimes without overlapping the test set.

| | |
|---|---|
| $\beta_{KL}$ | 0.1 |
| batch size | 8 |
| group size | 16 |
| $\delta$ | 0.95 |
| $\gamma$ | 0.95 |
| Rank | 64 |
| Alpha | 64 |
| LR | 3e-6 |
| Scheduler | Constant |
| Stage 1 Steps | 180 |
| Stage 2 Stage | 120 |

*Table A.9.* Hyperparameters used for training.

# P. Head-to-head Comparison with EditRetro

We visualize an intriguing head-to-head comparison of RetrOrchestrator with Retro*+EditRetro for planning the synthetic route of molecule A (COC[C@H](C)COCc1ccc([C@@]2(O)CCN(C(=O)OC(C)(C)C)C[C@@H]2C=NO)cc1) in Figure A.7 and Figure A.8. Both RetrOrchestrator and Retro*+EditRetro share the same first three reactions, leading to the intermediate molecule B (COC[C@H(C)COCc1ccc([C@@]2(O)CCN(C(=O)OC(C)(C)C)C[C@@H]2C(=O)O)cc1). At this point, the divergent model calls drastically change the overall outcome: RetrOrchestrator calls Graph2Edits (another editing-based SSR), while Retro*+EditRetro remains limited to the static choice of EditRetro. Graph2Edits predicts molecule C (COC[C@H(C)COCc1ccc([C@@]2(O)CCN(C(=O)OC(C)(C)C)C[C@@H]2C(=O)OC)cc1) and subsequently calls Chemformer, eventually finding a success route. In contrast, EditRetro predicts several complex molecules and fails to identify a successful path before the computational budget is exhausted. This demonstrates the critical advantage of RetrOrchestrator's adaptive model selection over static pipelines. By dynamically switching to Graph2Edits and subsequently Chemformer, RetrOrchestrator bypasses the dead ends that curses static systems. While the latter remains trapped by the limitations of a single predictive architecture, RetrOrchestrator leverages the diverse strengths of multiple SSR models to navigate complex chemical spaces more efficiently, ensuring high-quality route discovery even when initial candidates prove challenging.

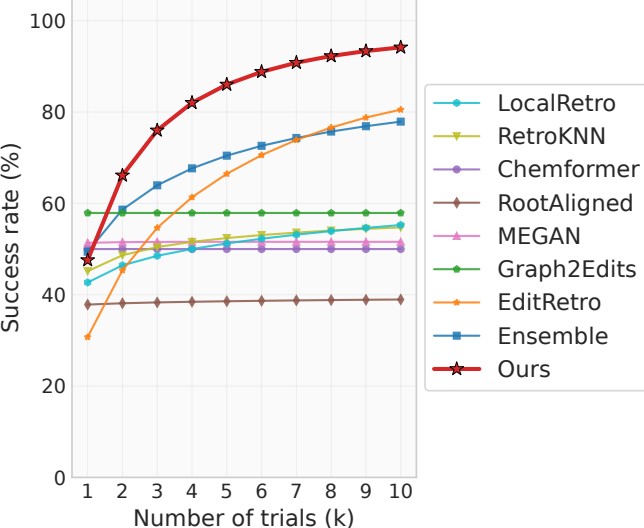

*Figure A.6.* Comparison of pass@k performance of RetrOrchestrator(ours) against various MDP baselines with Retro* planner.

# Q. Prompts

We use the prompt templates detailed below as input for the LLM agents. These templates contain dynamic fields enclosed in curly braces {}, which are populated by the environment. For the initial action in an episode, a template without conversational history is used; for all subsequent actions, the history is included.

Examples of textual observation and action history are shown in Example 1.

---

**Example 1: Template With Trajectory Histories**

You are an expert agent operating in the Retrosynthesis environment.
**Environment**
- You are operating in the Retrosynthesis environment.
- The environment is a retrosynthesis search tree, containing reaction nodes and molecule nodes.
- It's an AND-OR tree, where the AND nodes are the reaction nodes and the OR nodes are the molecule nodes.
- The root node is the target molecule.
- The tree will expand/update based on your actions.
**Your Goal**
Your goal is to synthesize the target molecule by iteratively expanding the synthesis tree using the given single-step models.
**Rules**
- You can only expand molecules using one of the given single-step models.
- You can only expand molecules that are in the tree and not yet expanded and known to be not purchasable.
- You can only terminate the search if a route with all leaf nodes in the retrosynthesis tree being purchasable is found.
- In each step, you should decide on what molecule to expand and which model to use. You can only query ONE model for ONE molecule at each step.
- Present your action between <action>...</action> tags and do not include any other text.
**Reward Structure**
- You receive a positive reward upon successful synthesis of the target molecule.
- Actions that do not lead to complete synthesis receive neutral rewards (zero), maintaining progress without premature termination.
**Actions Allowed**
- EXPAND <molecule_smiles> using <model_name> (predictions=<num_predictions>)
- TERMINATE search
**History**

---

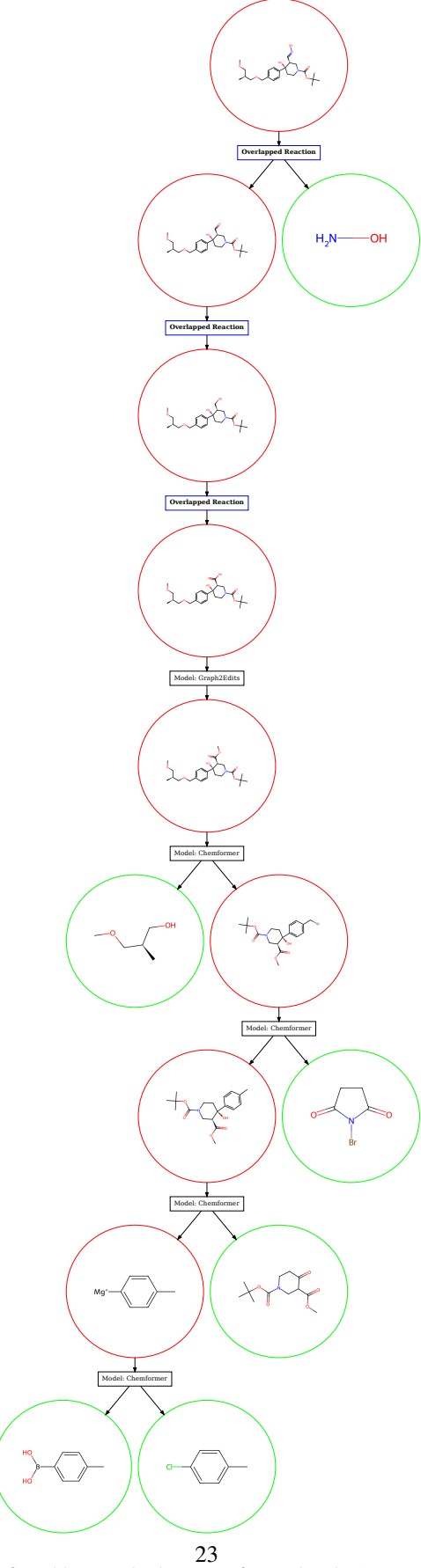

*Figure A.7.* Successful retrosynthesis route found by RetrOrchestrator for Molecule A. Reaction nodes labelled "Overlapped reaction" denote reactions that appear identically in the retrosynthesis routes produced by both Retro*+EditRetro and RetrOrchestrator.

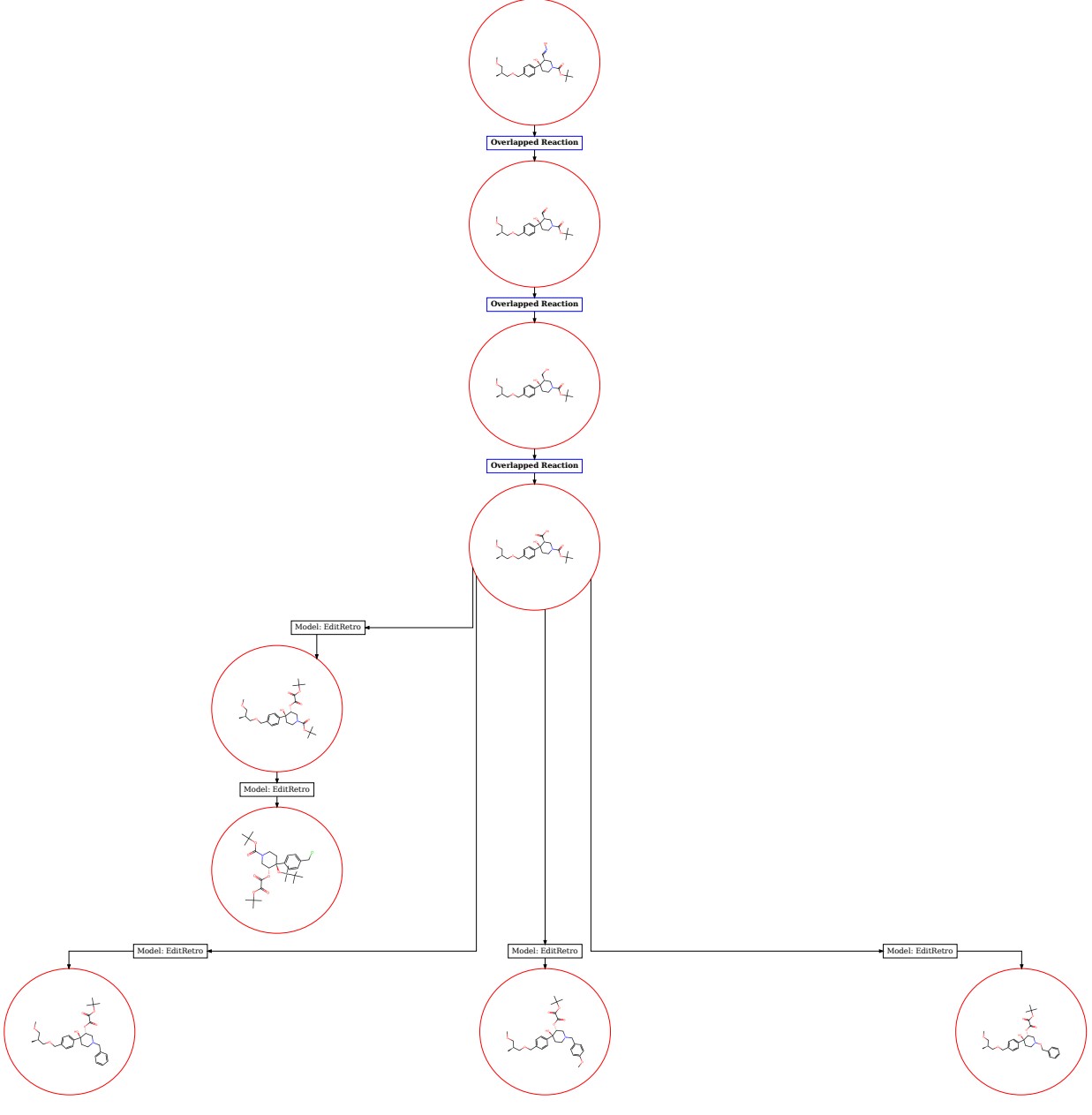

*Figure A.8.* Part of failed retrosynthesis route using Retro* and EditRetro for Molecule A. Reaction nodes labelled "Overlapped reaction" denote reactions that appear identically in the retrosynthesis routes produced by both Retro*+EditRetro and RetrOrchestrator.

{history}
**Current Observation**
{current_observation}
**Available Models**
{available_models}
**Available Actions**
TERMINATE search or EXPAND <any SMILES from expandable molecules> using <any model from AVAILABLE MODELS> (predictions=<number of predictions you think is best>).
It's your turn to take an action. Think carefully and then take a wise action.
Remember to wrap your action between <action>...</action> tags, DO NOT include any other text other than the action and the tags.

---

**Example 2: Template Without Trajectory Histories**

You are an expert agent operating in the Retrosynthesis environment.
**Environment**
- You are operating in the Retrosynthesis environment.
- The environment is a retrosynthesis search tree, containing reaction nodes and molecule nodes.
- It's an AND-OR tree, where the AND nodes are the reaction nodes and the OR nodes are the molecule nodes.
- The root node is the target molecule.
- The tree will expand/update based on your actions.
**Your Goal**
Your goal is to synthesize the target molecule by iteratively expanding the synthesis tree using the given single-step models.
**Rules**
- You can only expand molecules using one of the given single-step models.
- You can only expand molecules that are in the tree and not yet expanded and known to be not purchasable.
- You can only terminate the search if a route with all leaf nodes in the retrosynthesis tree being purchasable is found.
- In each step, you should decide on what molecule to expand and which model to use. You can only query ONE model for ONE molecule at each step.
- Present your action between <action>...</action> tags and do not include any other text.
**Reward Structure**
- You receive a positive reward upon successful synthesis of the target molecule.
- Actions that do not lead to complete synthesis receive neutral rewards (zero), maintaining progress without premature termination.
**Actions Allowed**
- EXPAND <molecule_smiles> using <model_name> (predictions=<num_predictions>)
- TERMINATE search
**Current Observation**
{current_observation}
**Available Models**
{available_models}
**Available Actions**
TERMINATE search or EXPAND <any SMILES from expandable molecules> using <any model from AVAILABLE MODELS> (predictions=<number of predictions you think is best>).
It's your turn to take an action. Think carefully and then take a wise action.
Remember to wrap your action between <action>...</action> tags, DO NOT include any other text other than the action and the tags.

**Example 3: Example of Observation**

=== RETROSYNTHESIS STATE (Step 1) ===
Target: `O=S(=O)(C#Cc1ccc(Cl)cc1)N1CCNCC1`
Status: UNSOLVED
Tree stats: 8 molecules, 5 reactions, depth 1
Budget remaining: 19
EXPANDABLE MOLECULES:

1. CC(C)(C)OC(=O)N1CCN(S(=O)(=O) C#Cc2ccc(Cl)cc2)CC1 (depth=1, visits=0)

2. O=S(=O)(Cl)C#Cc1ccc(Cl)cc1 (depth=1, visits=0)

3. O=S(=O)(C#Cc1ccc(Cl)cc1)N1CCN(Cc2ccccc2) CC1 (depth=1, visits=0)

4. C#CS(=O)(=O)N1CCNCC1 (depth=1, visits=0)

5. O=C(OCc1ccccc1)N1CCN(S(=O)(=O) C#Cc2ccc(Cl)cc2)CC1 (depth=1, visits=0)

SOLVED MOLECULES: 2 total

1. C1CNCCN1 (depth=1)

2. Clc1ccc(I)cc1 (depth=1)

CURRENT REWARD:
0.000
CUMULATIVE REWARD:
0.000
**Available Models**

- chemformer

- graph2edits

- localretro

- editretro

- megan

- retroknn

- root_aligned

**Example 4: Example of History**

History of the last 1 step:
=== RETROSYNTHESIS STATE (Step 1) ===
Target: `O=S(=O)(C#Cc1ccc(Cl)cc1)N1CCNCC1`
Status: UNSOLVED
Tree stats: 8 molecules, 5 reactions, depth 1
Budget remaining: 19
EXPANDABLE MOLECULES:

1. CC(C)(C)OC(=O)N1CCN(S(=O)(=O) C#Cc2ccc(Cl)cc2)CC1 (depth=1, visits=0)

2. O=S(=O)(Cl)C#Cc1ccc(Cl)cc1 (depth=1, visits=0)

3. O=S(=O)(C#Cc1ccc(Cl)cc1)N1CCN(Cc2ccccc2) CC1 (depth=1, visits=0)

4. C#CS(=O)(=O)N1CCNCC1 (depth=1, visits=0)

5. O=C(OCc1ccccc1)N1CCN(S(=O)(=O) C#Cc2ccc(Cl)cc2)CC1 (depth=1, visits=0)

SOLVED MOLECULES: 2 total

1. C1CNCCN1 (depth=1)

2. Clc1ccc(I)cc1 (depth=1)

CURRENT REWARD:
0.000
CUMULATIVE REWARD:
0.000
Action:
EXPAND O=S(=O)(C#Cc1ccc(Cl)cc1)N1CCNCC1 using graph2edits (predictions=5)
Reward:
0
Result: {
'success': True,
'action_type': 'expand_molecule',
'target_smiles':
'O=S(=O)(C#Cc1ccc(Cl)cc1)N1CCNCC1',
'model_name': 'graph2edits',
'num_predictions': 5,
'predictions':
'Top 3 predictions:
CC(C)(C)OC(=O)N1CCN(S(=O)(=O)C#Cc2ccc (Cl)cc2)CC1 (prob: 0.646)
C1CNCCN1.O=S(=O)(Cl)C#Cc1ccc(Cl)cc1 (prob: 0.352)
O=S(=O)(C#Cc1ccc(Cl)cc1)N1CCN(Cc2ccccc2)CC1 (prob: 0.002) '
}

# R. Clustering of Retro*-190 Molecules

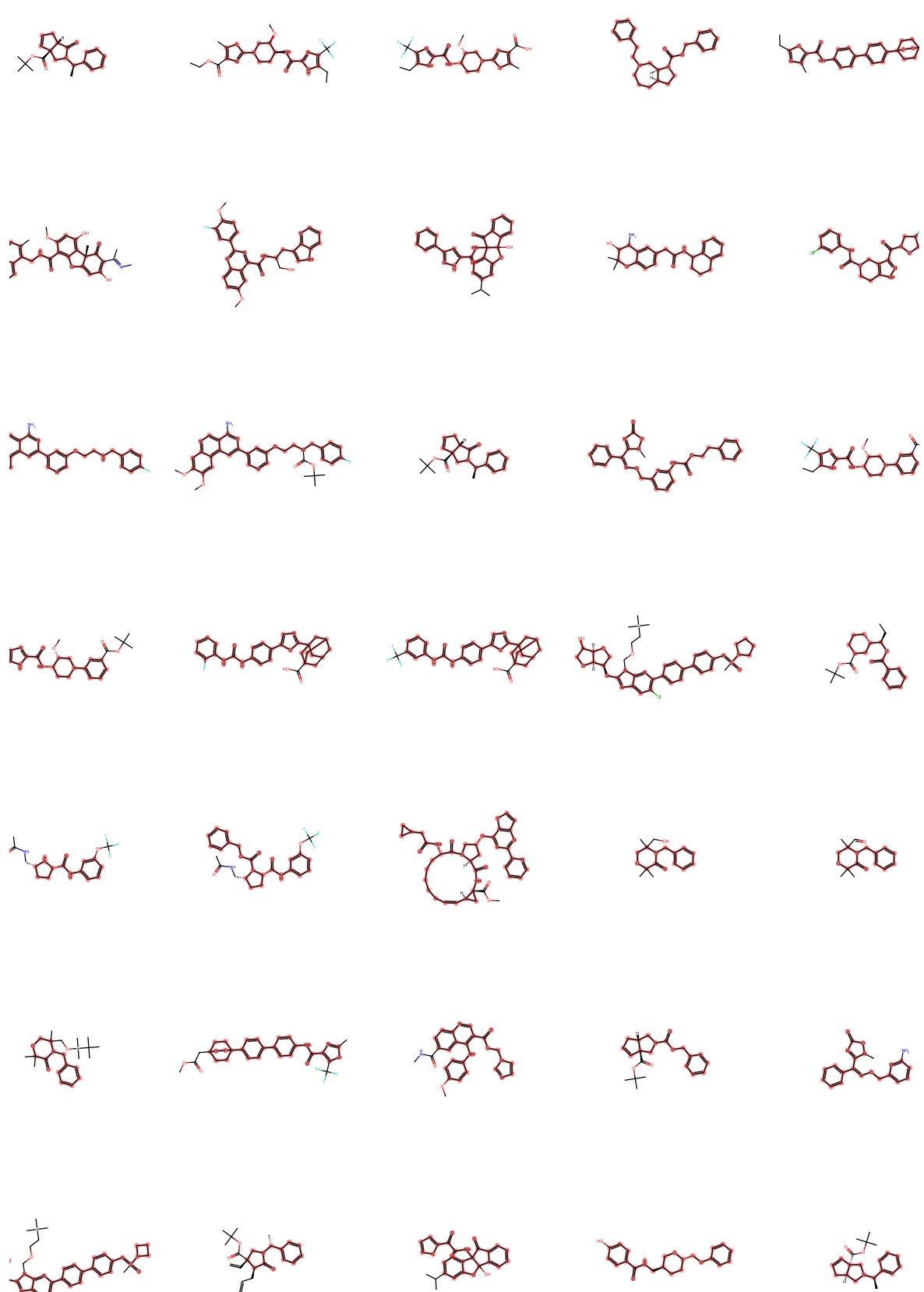

*Figure A.9.* Molecules in scaffold cluster 1 (part 1/2)

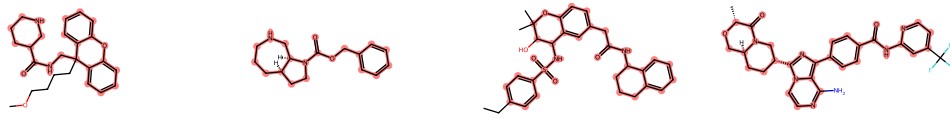

*Figure A.10.* Molecules in scaffold cluster 1 (part 2/2)

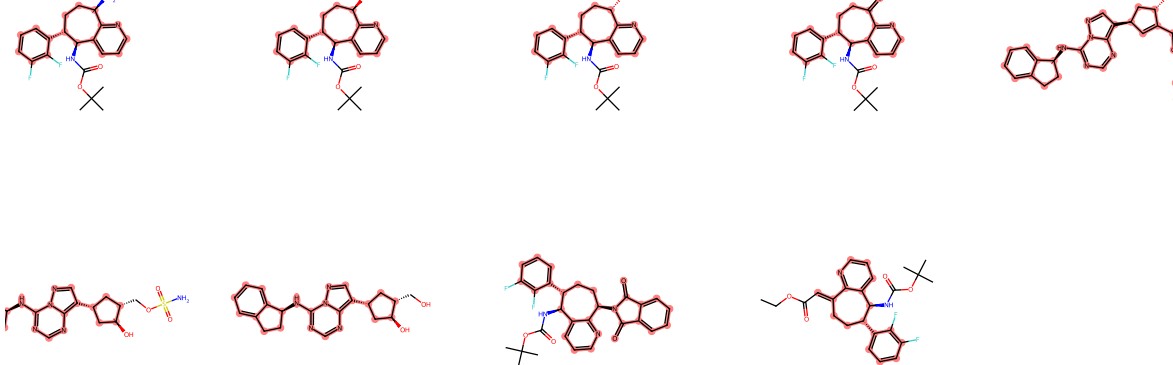

*Figure A.11.* Molecules in scaffold cluster 2

| **LocalRetro** | |
| --- | --- |
| bound_constant | 1 |
| policy_kwargs.temperature | 4.0 |
| value_function_kwargs.constant | 0.5 |
| **RetroKNN** | |
| bound_constant | 1 |
| policy_kwargs.temperature | 8.0 |
| value_function_kwargs.constant | 0.75 |
| **Chemformer** | |
| bound_constant | 1 |
| policy_kwargs.temperature | 8.0 |
| value_function_kwargs.constant | 0.75 |
| **Root Aligned** | |
| bound_constant | 10 |
| policy_kwargs.clip_probability_max | 0.999 |
| policy_kwargs.clip_probability_min | 1.0e-05 |
| policy_kwargs.temperature | 8.0 |
| value_function_kwargs.constant | 0.5 |
| **MEGAN** | |
| bound_constant | 1 |
| policy_kwargs.clip_probability_max | 0.9999 |
| policy_kwargs.clip_probability_min | 1.0e-05 |
| policy_kwargs.temperature | 2.0 |
| value_function_kwargs.constant | 0.75 |
| **Graph2Edits** | |
| bound_constant | 1 |
| policy_kwargs.temperature | 4.0 |
| value_function_kwargs.constant | 0.5 |
| **EditRetro** | |
| bound_constant | 1 |
| policy_kwargs.temperature | 1.0 |
| value_function_kwargs.constant | 0.5 |
| **Ensemble** | |
| bound_constant | 1 |
| policy_kwargs.temperature | 1.0 |
| value_function_kwargs.constant | 0.5 |

*Table A.10.* Tuned hyper-parameters settings for MCTS baselines.

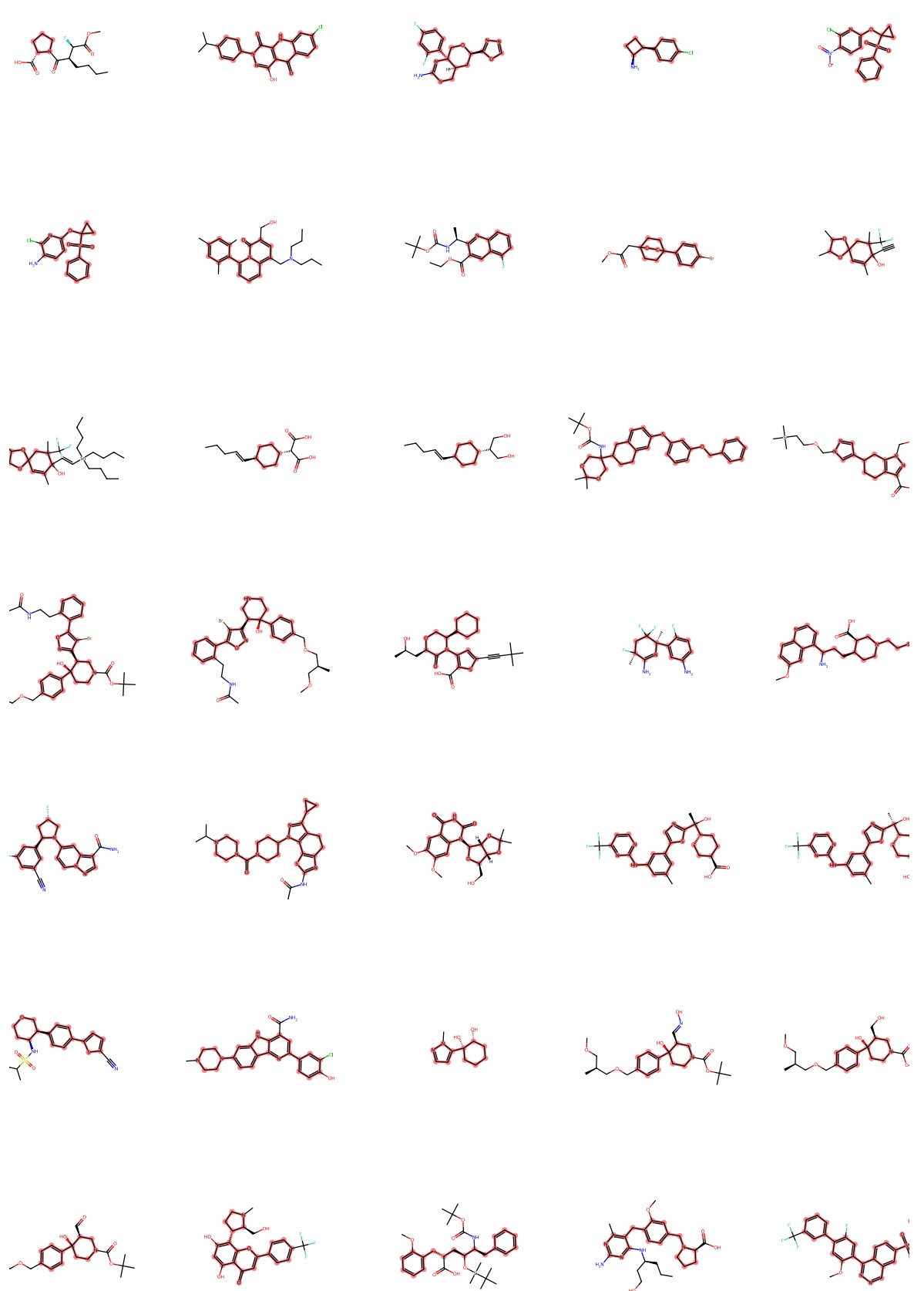

*Figure A.12.* Molecules in scaffold cluster 3 (part 1/3)

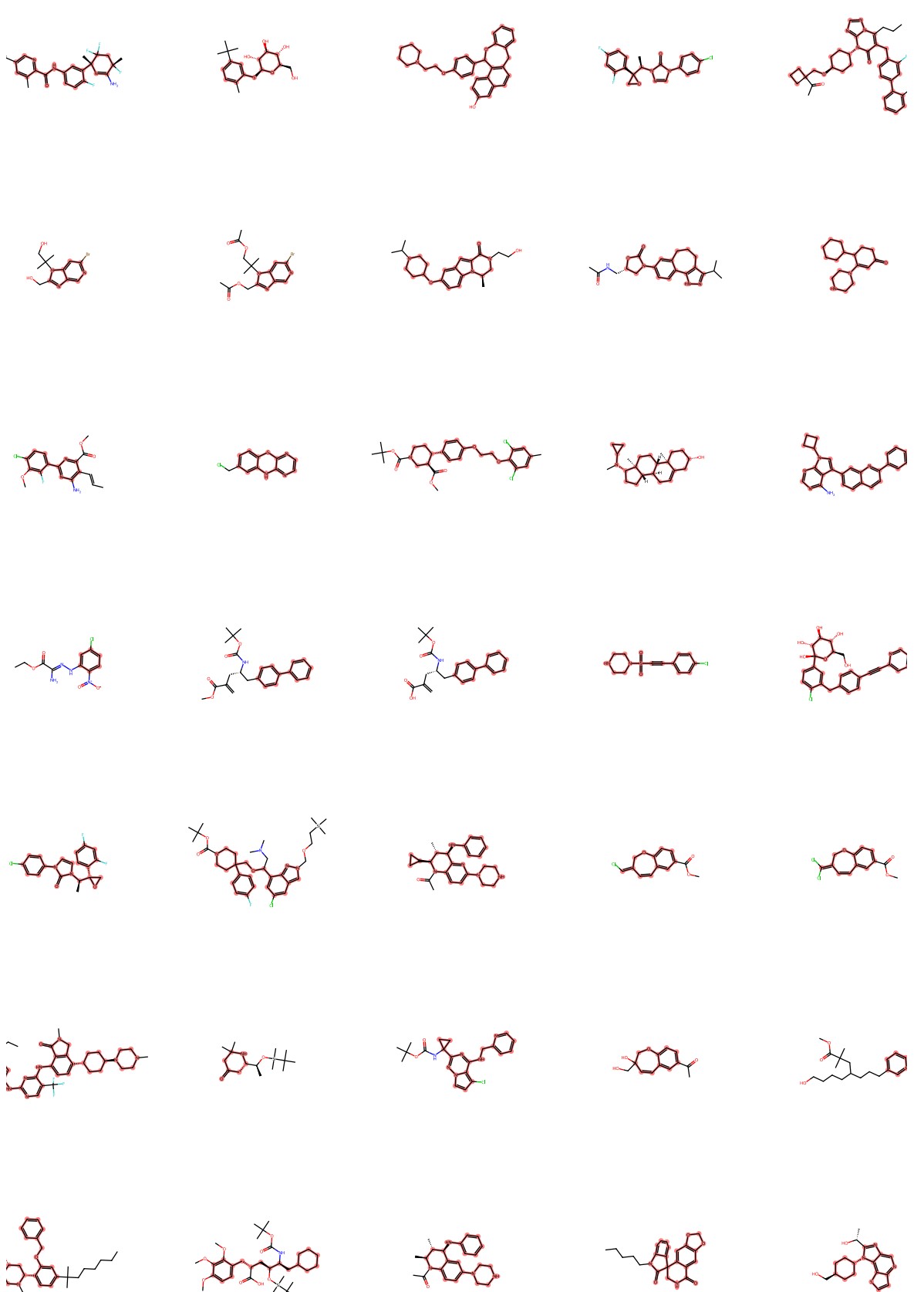

*Figure A.13.* Molecules in scaffold cluster 3 (part 2/3)

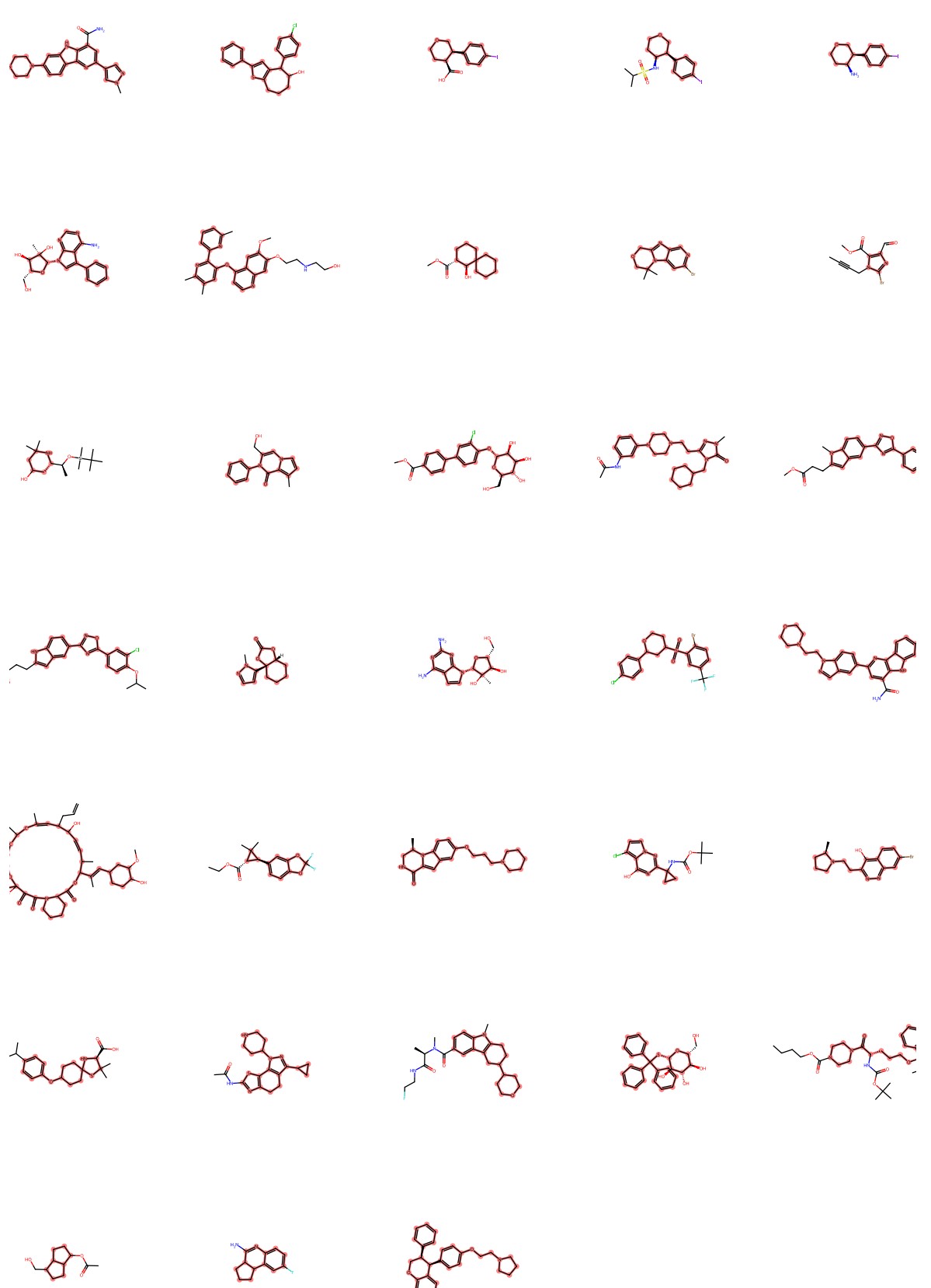

*Figure A.14.* Molecules in scaffold cluster 3 (part 3/3)

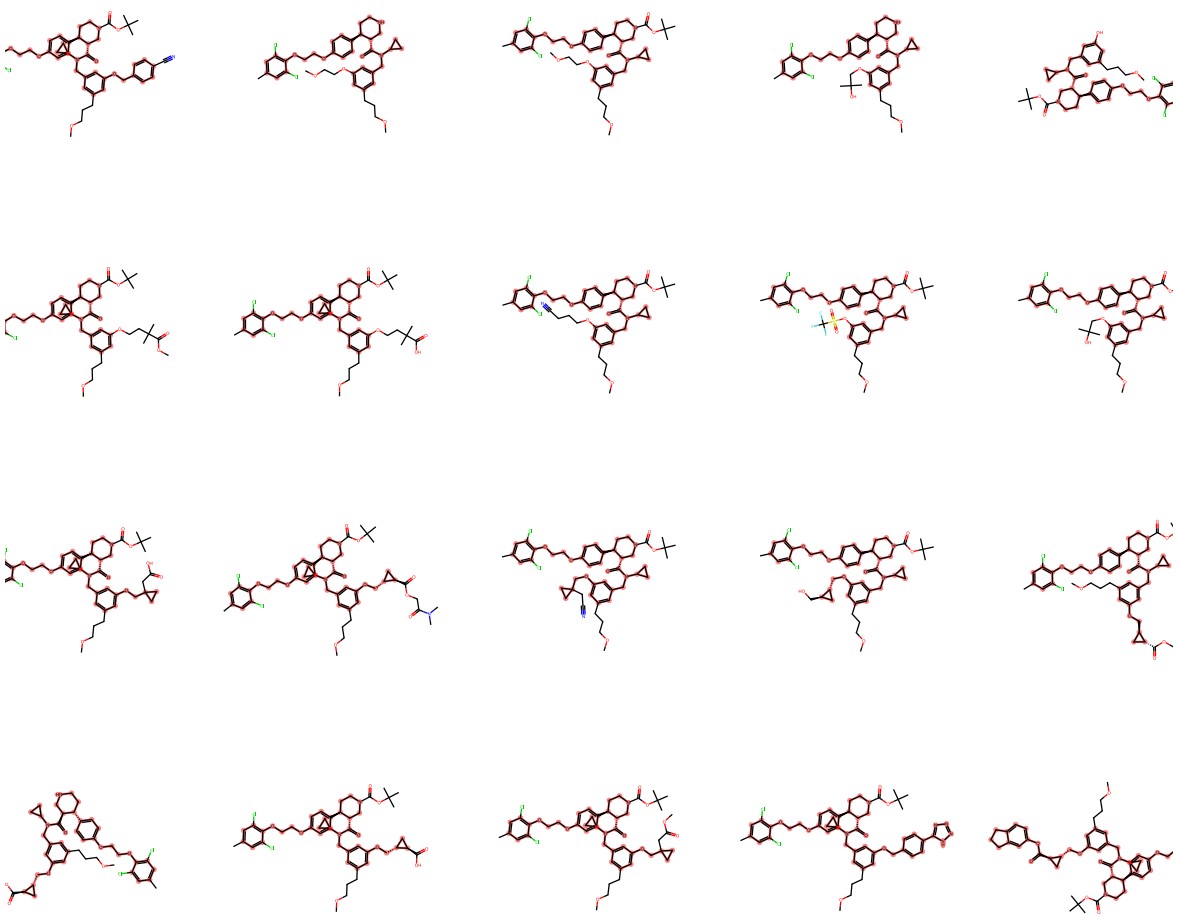

*Figure A.15.* Molecules in scaffold cluster 4

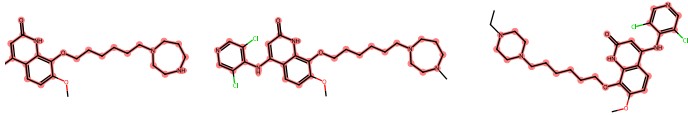

*Figure A.16.* Molecules in scaffold cluster 5

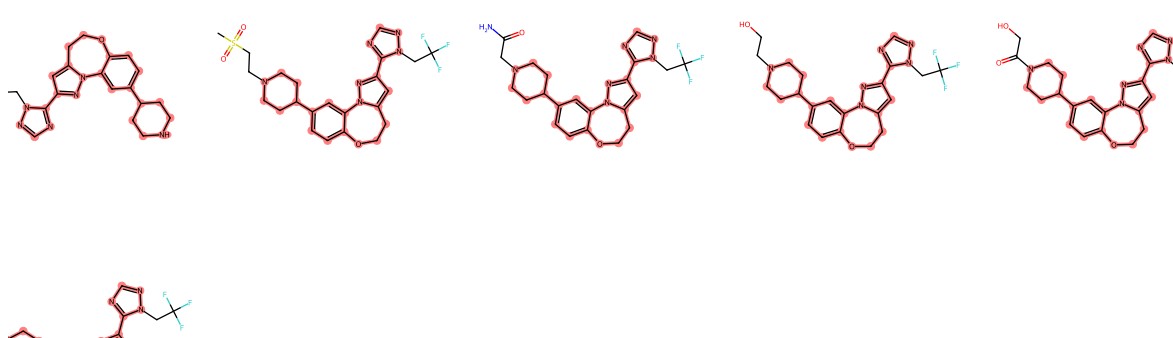

*Figure A.17.* Molecules in scaffold cluster 6

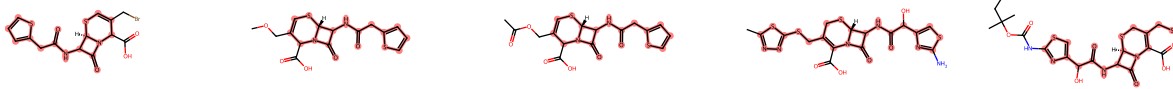

*Figure A.18.* Molecules in scaffold cluster 7

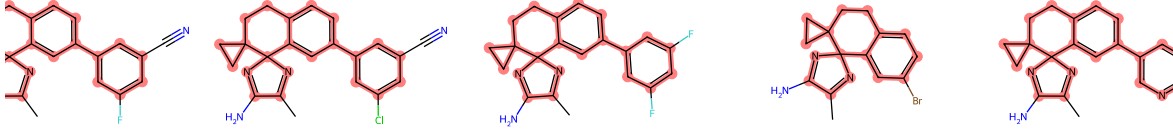

*Figure A.19.* Molecules in scaffold cluster 8

