# OpenReview forum: "RetrOrchestrator: A Multi-Step Retrosynthesis Agent Dynamically Orchestrating Single-Step Transition Models"
_ICML.cc/2026/Conference — ICML 2026 regular_

### Official Review · Reviewer_Gnxp · 2026-02-24

**Soundness:** 3
**Presentation:** 3
**Significance:** 2
**Originality:** 2
**Overall Recommendation:** 4
**Confidence:** 4

**Summary:**

This paper introduces RetrOrchestrator, an LLM-powered agent that explicitly accounts for the model skill disparity among multiple single-step retrosynthesis models. It proposes SA-GRPO, which leverages clustering to provide improved training signals for fine-tuning the LLM to select appropriate single-step actions. Experimental results on the Retro*-190 benchmark demonstrate the superiority of the proposed method from multiple perspectives.

**Compliance With Llm Reviewing Policy:**

Affirmed.

**Final Justification:**

The authors' response to Q4 is clearly articulated, and it is recommended that they incorporate it into the final manuscript. The explanation provided for Q5 is fine, and no further concerns need to be pursued on this point. Overall, this work is reasonably complete, and I have revised my decision to a weak accept. However, considering the novelty, I maintain a neutral position and would also not strongly advocate for acceptance.

**Key Questions For Authors:**

1. A special TERMINATE action is included to end the search. Why is this action necessary, given that prior work such as Retro* does not explicitly include such an action?

2. The paper states that any invalid action incurs a penalty of −0.1. How is an action determined to be invalid? Does this definition account for cases where the reaction proposed by the SSR model is chemically infeasible? In the experiments reported in Table 1, do the discovered routes include invalid actions during the search process?

3. The clustering groups structurally similar molecules together, implicitly assuming that similar structures share similar retrosynthesis pathways. However, this assumption may not always hold. For example, for two structurally similar molecules A and B, the precursors for synthesizing A may happen to be available commercially, while those for B may not be. In addition, which set of commercially available molecules is used in the experiments?

4. The final advantage $A_{i,t}$ is defined as the sum of $A_i^{E}$ and $A_{i,t}^{S}$ with equal contribution. Why is it still necessary to retain $A_i^{E}$? How does the performance change when different weights are assigned to these two terms?

5. Regarding the reported success rate and pass@10, the improvement may be partly due to increased randomness in the generated actions, which can artificially boost pass@10. Could the authors also report pass@1 results to better reflect the intrinsic capability of the model?

6. Since line 173 states that the number of reactions to generate is part of the action space learned by the LLM, why is the number of reaction predictions returned per model per call still capped at 10 (as mentioned in line 376)?

**Limitations:**

I do not identify any significant potential negative societal impact associated with this work.

**Strengths And Weaknesses:**

**Strengths**
1. This paper is clearly written and easy to follow.

2. It considers SSA selection from a new perspective by dynamically choosing among multiple single-step models, which is an interesting angle.

3. The paper compares against a wide range of baseline methods, which strengthens the empirical evidence and makes the results more convincing.

**Weaknesses**

1. The experiments could be more comprehensive, as certain aspects do not fully demonstrate the effectiveness of the proposed method and its improvements (see the questions section for details).

2. The work is primarily engineering-focused, and the algorithmic novelty appears somewhat incremental.

---

> ### Author Rebuttal · Authors · 2026-03-31
>
> We sincerely thank Reviewer Gnxp. Additional rebuttal results are at https://anonymous.4open.science/r/RetrOrchestratorRebuttal-FCC3; rebuttal tables and figures are cited as Table/Fig. rX.
>
> **[W1] Comment on algorithmic novelty**
> - We clarify **our conceptual and algorithmic novelty**, beyond engineering. Our work exposes a fundamental overlooked challenge in retrosynthesis planning: the **uni-model transition assumption**. Because SSRs exhibit clear skill disparity, using one fixed SSR across all planning steps is suboptimal.
> - We therefore reformulate planning from an MDP to a **POMDP in which transition-model selection is explicitly included in the learned policy**.
> - This is **not captured by straightforward extensions of existing methods**:
>   - a naive SSR router reaches only **82.63% vs 94.21%** (Table r7);
>   - an off-the-shelf LLM planner without RL reaches only **9.47%** (Table 1);
>   - standard RL is insufficient because **sequence-level advantage fails to account for skill disparity**.
> - We therefore introduce **scaffold-aware RL** and a **molecule curriculum**.
>
> **[Q1] TERMINATE action in ours**
> - The need for TERMINATE comes from **the difference in search behavior between conventional and LLM planners**.
> - In conventional planners such as Retro*, **stopping is handled by an external controller**; the planner is only used to select molecules.
> - In contrast, our setting follows **agentic sequential decision making [1][2]**, where the **policy is fully responsible for the planning process**. Thus, "whether to continue" must be represented in the action space.
> - TERMINATE is valid only when a full route with purchasable leaves has been found; otherwise it is invalid and penalized.
>
> **[Q2] Invalid action definition**
> - As shown by the prompt example in Line 951, an action is invalid if it **cannot be executed by the planning environment**, including malformed / unparsable output, selecting a non-expandable molecule, choosing an unavailable SSR or illegal prediction count, invoking an SSR that returns no expansion, or issuing TERMINATE before a valid route is found.
> - We **do take chemical feasibility into account** in the fourth case: if the proposed reactions are chemically infeasible, the SSR returns no valid expansion. Following [3], we filter infeasible reactions using the confidence thresholds in Table A.4.
> - Invalid actions may appear during training or search and are penalized, but since they do not modify the synthesis graph, **final discovered routes contain no invalid action**.
>
> **[Q3] Assumption that similar molecules share similar pathways**
> - We respectfully clarify that **our method does not rely on structurally similar molecules sharing identical retrosynthetic routes**.
> - Scaffold clustering is introduced only to construct a **lower-variance baseline for step-level credit assignment** (Eqn. 9).
> - This step-level signal is still **grounded on route-level reward** (Eqn. 8). When precursor availability differs, route-level rewards naturally diverge.
> - This is why we use a relatively strict threshold **$\delta=0.95$** and retain the episodic advantage term.
> - For all experiments, we use Retro*'s **23.1M** purchasable-molecule inventory.
>
> **[Q4] Ablating $A_i^E$ and $A_{i,t}^S$ weights**
> - We retain both terms because they are **complementary**: **$A_i^E$** aligns with final route success, while **$A_{i,t}^S$** provides dense local credit.
> - Table r4 shows: episodic only **74.21%**, step-level only **72.22%**, equal weights **94.21%**, and over-weighting the step-level term ($A_i^E + 5A_{i,t}^S$) degrades performance to **60.00%**.
> - Thus, **simple summation works**, consistent with [4].
>
> **[Q5] Pass@1 results**
> - We agree pass@1 is important, and we **have included the pass@k plot in the appendix**.
> - Although RetrOrchestrator is not the best at pass@1, its gain appears already at **pass@2** and remains above all static baselines afterward.
> - This matches our setting: the method's strength is **dynamically reallocating search effort after initial interaction**, rather than maximizing a single fixed first-shot ranking, so we do not believe the gain is explained by large-$k$ randomness alone.
>
> **[Q6] Cap on number of reactions generated**
> - A cap on the branching factor **is not in conflict with** modeling it as part of the action space; it defines a practical bounded range.
> - Within this range, the policy still adjusts branching factor based on the state. Table r8 shows a positive correlation with search phase (Spearman **0.211**): fewer candidates early, more later as search becomes harder.
> - The cap of **10** is also **not arbitrary**. Prior work [3] shows rapid early gains, a practical sweet spot around 5, much slower growth beyond 10, and little additional gain beyond 50.
>
> We thank the reviewer again.
>
> [1] Nakano et al., WebGPT.
> [2] Yao et al., WebShop.
> [3] Maziarz et al., Re-evaluating retrosynthesis algorithms with syntheseus.
> [4] Liu, Shih-Yang, et al., GDPO.

---

> > ### Author Rebuttal · Reviewer_Gnxp · 2026-04-03
> >
> > Thank you for the clarification. I still have the following concerns. Could the authors explain in more detail why $A^E$ and $A^S$ should be regarded as \emph{complementary} rather than largely \emph{correlated}, given that both are ultimately derived from trajectory-level returns? In addition, it would be helpful if the authors could provide a more detailed ablation study on the combination $A^E + \lambda A^S$. In particular, please show whether the method remains robust over a reasonable range of $\lambda$, rather than relying on a specific parameter choice. Such an analysis would help clarify whether the reported gains reflect a stable property of the proposed design or are partly due to parameter tuning. Moreover, the observation that the performance is degraded to $60.00\\%$ when $\lambda = 5.0$ further suggests that the interaction between $A^E$ and $A^S$ is not straightforward. This result is not only worse than $\lambda = 1$, but also worse than the $\lambda = \infty$ setting (i.e., step-level only, $72.22\\%$). If $A^E$ and $A^S$ were truly complementary in a simple sense, such behavior would be difficult to explain. A more detailed analysis is needed to understand why a heavily weighted combination can underperform the step-level-only variant, and whether this indicates optimization instability, or a more fundamental inconsistency between the two advantage terms.
> >
> > Since pass@1 directly evaluates the quality of the \emph{guiding policy} in finding the first successful solution. In essence, a strong policy should maximize the probability of identifying a good solution with as few interactions as possible, ideally within a single attempt. From this perspective, pass@1 is arguably the most direct indicator of whether the learned policy is truly effective. Therefore, if the proposed method does not perform well on pass@1 and instead relies on additional interactions to reallocate search effort, this raises the question of whether the learned policy is sufficiently strong in the first place. Put differently, does the need for multiple rounds of interaction suggest that the policy is not yet able to guide the search effectively at the earliest stage? Does this mean that the algorithm performs poorly at reallocating search effort before finding the first solution?

---

> > > ### Author Response · Authors · 2026-04-08
> > >
> > > We sincerely appreciate the reviewer's prompt follow-up and the opportunity for us to further clarify the remaining concerns.
> > > We updated our anonymous repository ([link](https://anonymous.4open.science/r/RetrOrchestratorRebuttal-FCC3/)) with the new ablation results.
> > >
> > > ### **[Q4] Why are $A^E$ and $A^S$ complementary rather than merely correlated?**
> > > > - We respectfully clarify that by "complementarity", we mean both $A^E$ and $A^S$ are **indispensable**. This **does not contradict** their inherent **correlation** stemming from trajectory-level returns. We will revise the text.
> > > >    - **$A^E$** evaluates rollout against a **batch-level baseline**, serving as a **global trajectory anchor**;
> > > >    - **$A^S$** compares within **scaffold-similar states**, providing **local state-conditional credit**.
> > > >    - Pairing each step-level $A^S$ with its rollout-level $A^E$ yields Pearson $r=0.537$ and Spearman $\rho=0.469$, confirming they **capture distinct complementary signals despite being correlated**.
> > > >
> > > > - Their interaction behavior **aligns with both their complementary and correlated nature**.
> > > >   - Following the reviewer's suggestion, we ablated $\lambda$ in $A = A^E + \lambda A^S$. Due to compute and time limits, these are evaluated after **60 Stage-1 steps**, but the trend remains consistent:
> > > >      - A **clear rise-then-fall trend**: performance peaks at **$\lambda=1.0$**, remains strong for $\lambda \in [0.5,2.0]$, and degrades when $A^S$ is over-weighted.
> > > >      - **Step-only outperforms $\lambda=5.0$**: we will explain this behavior below.
> > > >
> > > >|$\lambda$|0.0|0.5|1.0|1.5|2.0|2.5|5.0|Step Only|
> > > >|---|---:|---:|---:|---:|---:|---:|---:|---:|
> > > >|Success rate (%)|42.63|48.95|63.16|58.95|49.47|42.11|34.21|41.57|
> > > >
> > > >    -  We humbly clarify that **the Step-only setting is not equivalent to taking $\lambda\rightarrow \infty$, thus it does not contradict the above interaction behavior**.
> > > >       - Mathematically, the policy-gradient is proportional to $-(A^E+\lambda A^S)is(\theta)\nabla_\theta \log \pi_\theta(o_t|s_t)$ where $is(\theta)$ is the importance sampling ratio being clipped. Increasing $\lambda$ amplifies $\lambda A^S$, causing larger gradients and destabilized optimization.
> > > >         - We validate this empirically (**Fig. r5**, **Table r12**). The average gradient norm is **0.101** (Step-only), **0.099** ($\lambda=1.0$), but jumps to **0.324** ($\lambda=5.0$).
> > > >         - Thus, large $\lambda$ induces much larger policy updates and destabilizes the training process, explaining the degradation. In contrast, **Step-only avoids this scalar amplification with unweighted $A^S$**, preventing such instability.
> > >
> > > ### **[Q5] Why is pass@1 not the only relevant metric here?**
> > > > - We fully agree on the primary importance of pass@1 for measuring single-run success probability.
> > > > - However, we respectfully clarify **why pass@$k$** (repeated-run success coverage) **also matters critically here**.
> > > >   - First, our planner is inherently stochastic (like LocalRetro/EditRetro, Fig. A.2), yielding varying outcomes across runs.
> > > >      - **In such stochastic environments**, [1] defines policy quality as success across multiple runs under uncertainty, making pass@$kz$ an important metric [2]. Notably, relative performance between pass@1 and pass@$k$ is not fixed [3].
> > > >      - Our planner achieves the highest pass@1 among stochastic planners.
> > > >   - Second, **retrosynthesis planning is a domain where diversity is intrinsically important**: molecules admit multiple valid routes, and providing diverse pathways is critical [4,5]. Furthermore, route quality is often debated [4], and computationally valid routes may lack wet-lab feasibility [6] (e.g., unoptimized low yields).
> > > >      - Accordingly, we report pass@$k$ and **Avg. No. of Unique Routes** (Table 1). Compared to methods with stronger **pass@1**, ours achieves higher diversity and pass@$k$, demonstrating its **superiority in producing diverse and high-quality pathways**.
> > >
> > > We hope these clarifications address the remaining concerns, and we would be grateful if the reviewer could take these additional analyses into account in the final assessment.
> > >
> > > [1] Kaelbling et al. *Planning and Acting in Partially Observable Stochastic Domains*. Artificial Intelligence, 1998.
> > > [2] Chen et al. *Evaluating Large Language Models Trained on Code*. Technical report, 2021.
> > > [3] Gehring et al. *RLEF: Grounding Code LLMs in Execution Feedback with Reinforcement Learning*. ICML'25.
> > > [4] Franz, C., et al. *Completeness and Diversity in Depth-First Proof-Number Search with Applications to Retrosynthesis.* IJCAI'22.
> > > [5] Zhao, et al. *A comprehensive survey of AI-based retrosynthesis planning: Datasets, models, and tools.* The Innovation Informatics (2025): 100026-1.
> > > [6] Maziarz, K., et al. *Re-evaluating retrosynthesis algorithms with syntheseus.* Faraday Discussions 256 (2025): 568-586.

---

### Official Review · Reviewer_NWE3 · 2026-03-12

**Soundness:** 3
**Presentation:** 3
**Significance:** 3
**Originality:** 3
**Overall Recommendation:** 5
**Confidence:** 3

**Summary:**

This paper introduces RetrOrchestrator, an LLM-powered agent for multi-step retrosynthesis planning that dynamically selects which single-step retrosynthesis (SSR) model to apply at each expansion step. The key observation motivating the work is that different SSR models (template-based, template-free, editing-based) exhibit pronounced "skill disparity" — no single model dominates across all molecular scaffolds. Rather than committing to a fixed SSR as in prior MDP-based planners, the authors reframe the problem as a Partially Observable Markov Decision Process (POMDP), where the LLM agent's KV cache serves as an implicit belief state that evolves with search feedback.

The agent is trained with a novel reinforcement learning algorithm called Scaffold-Aware Group Relative Policy Optimization (SA-GRPO). SA-GRPO addresses the sparse reward problem of trajectory-level signals by introducing a step-level advantage computed over clusters of intermediate observations grouped by Bemis-Murcko scaffold similarity. Training follows a two-stage curriculum: Stage 1 teaches chemical rule adherence on easier molecules (SCScore < 4), while Stage 2 focuses on strategic planning on harder molecules (SCScore > 4).

On the Retro\*-190 benchmark, RetrOrchestrator achieves a 94.21% success rate (built on Qwen2.5-1.5B), surpassing the previous best LLM-based planner (90.53%, backed by DeepSeek-R1 with 671B parameters) and all reported static MDP baselines.

**Compliance With Llm Reviewing Policy:**

Affirmed.

**Final Justification:**

The code was released (although I couldn't check it, at least it is there). I raised my score.

**Key Questions For Authors:**

1. **Ensemble budget fairness:** How exactly are expansion calls counted for the Ensemble baseline? If one expansion equals one collective call to all 7 SSR models, the ensemble uses 7× more model queries per expansion than RetrOrchestrator's single-model calls. Please report the total number of individual SSR model calls for the ensemble to enable a fair efficiency comparison. A clarification here could change my assessment of the efficiency claims.

2. **Statistical reliability:** Given the small test set (190 molecules), the 3.7 percentage point improvement over Song et al. (2025b) corresponds to ~7 molecules. Can you provide confidence intervals or a significance test (e.g., McNemar's test) for the success rate comparison? This would affect my confidence in the headline result.

3. **Failure analysis:** What are the characteristics of the ~11 molecules that RetrOrchestrator fails to solve? Are these cases where all individual SSR models also fail, or are there cases where a static baseline succeeds but the agent's routing decisions lead to failure? This analysis would clarify the method's ceiling and potential failure modes.

4. **Curriculum sensitivity:** How sensitive is performance to the SCScore = 4 threshold and the ratio of Stage 1 (180 steps) to Stage 2 (120 steps) training? An ablation here would strengthen the claims about the curriculum's contribution.

5. **Scalability and generalization:** The OOD evaluation on PDB-120 shows a 65.83% success rate, which is a notable drop from 94.21% on Retro\*-190. What drives this gap — is it the distributional shift in molecular complexity, or are some SSR models less reliable on PDB-type ligands? This could inform how the method might be extended to larger and more diverse chemical spaces.

**Limitations:**

- The still-significant wall-clock time overhead introduced by LLM inference, which may limit practical deployment in high-throughput settings.
- The reliance on a fixed set of 7 SSR models — the framework's performance is upper-bounded by the collective capabilities of these models.
- The small benchmark sizes and the need for validation on larger, more diverse molecular datasets.
- The fact that all SSR models are trained on USPTO-50K, which itself has known biases and limited chemical diversity.

**Strengths And Weaknesses:**

### Strengths

- **Well-motivated problem formulation.** The empirical observation that SSR models exhibit scaffold-dependent skill disparity (Figure 1) is compelling and clearly establishes the need for dynamic model selection. The radar plot effectively communicates that no single SSR is universally best.

- **Principled POMDP formulation.** Reframing retrosynthesis planning from MDP to POMDP to account for model selection uncertainty is a clean conceptual contribution. The connection between the LLM's KV cache and the POMDP belief state is well-motivated by recent theoretical work (Piotrowski et al., 2025).

- **Strong empirical results.** Achieving 94.21% success rate with a 1.5B parameter model, outperforming a system backed by a 671B parameter LLM, is great. The ablation cascade (9.5% → 65.4% → 74.2% → 94.2%) clearly demonstrates the contribution of each component.

- **Pareto efficiency analysis.** Figure 6 convincingly shows that RetrOrchestrator achieves superior success rates with roughly half the model calls of many baselines, establishing a meaningful efficiency–performance frontier.

- **Interesting emergent behaviors.** The analysis of how the agent's model selection policy evolves from feature-driven (Stage 1) to performance-aligned (Stage 2) routing is insightful (Figure 5), and the cluster-specific deviations (e.g., favoring Chemformer for Cluster 5) demonstrate genuine adaptive behavior beyond simply learning to always call the best overall model.

- **Head-to-head case study.** The comparison in Appendix H, where RetrOrchestrator and Retro\*+EditRetro share the first three steps but diverge in outcome, is an effective qualitative demonstration of the method's advantage.

### Weaknesses

- **Code availability and reproducibility.** The paper does not indicate whether code will be released. Given the complexity of the system (LLM agent, 7 SSR model backends, custom RL training pipeline built on verl-agent), reproducibility is a significant concern. Will the code and trained models be made available? Under what license?

- **Narrow benchmark evaluation.** The primary evaluation is on Retro\*-190, which contains only 190 molecules. While the OOD evaluation on PDB-120 is appreciated, both benchmarks are quite small. The statistical reliability of percentage differences on 190 molecules should be discussed more carefully. For instance, the 3.7 percentage point improvement over Song et al. (2025b) corresponds to roughly 7 additional molecules solved — how robust is this to random seed variation?

- **Incomplete comparison to ensemble baselines.** The Ensemble baseline uses all SSR models jointly at each expansion step, but the paper is not entirely clear on how the expansion budget is counted for the ensemble. Table 1 shows the ensemble achieving only 4.48 average expansions with Retro\*, which seems very low. How are calls counted for the ensemble — is one collective call to all 7 models counted as a single expansion? If so, the ensemble uses 7× more model queries per expansion. This makes the efficiency comparison in Figure 6 potentially misleading. The number of total SSR model calls (not just expansions) for the ensemble should be explicitly reported.

- **Incomplete comparison to most recent methods on the Retro\* benchmark.** Retrosynthetic Planning with Dual Value Networks. Liu et al., Proceedings of the 40th International Conference on Machine Learning, PMLR 202:22266-22276, 2023, report a stronger baseline than the one reported. Under a computational budget of 100, they report a 96.84% success rate. How would it perform in the Pareto front analysis?

- **Wall-clock time analysis is incomplete.** RetrOrchestrator's average wall-clock time of 118.9s is not negligible and is substantially higher than many individual SSR baselines (e.g., Retro\*+LocalRetro at 1.85s, Retro\*+Graph2Edits at 17.1s). The paper acknowledges LLM latency but does not break down where time is spent (LLM inference vs. SSR model calls vs. environment overhead). This decomposition would be valuable for understanding practical bottlenecks.

- **Limited analysis of failure cases.** 94.21% means roughly 11 molecules out of 190 are still not solved. What characterizes these failures? Are they molecules where no SSR model succeeds individually, or cases where the agent makes suboptimal routing decisions? This analysis would clarify the method's ceiling.

- **The POMDP formulation, while elegant, may overstate the theoretical contribution.** The belief state is never explicitly maintained or reasoned about — it is simply the LLM's hidden state, which is standard for any autoregressive LLM agent with a context window. The POMDP framing provides conceptual clarity but the practical mechanism (LLM reads observation history, generates an action) is identical to what any prompted LLM agent would do.

- **Training data leakage concerns.** The Retro\*-190 test molecules and the Stage 1/Stage 2 training molecules are both drawn from datasets associated with Retro\*. While the authors state that training uses the Retro\* training set, it would strengthen the paper to explicitly confirm there is no overlap between training molecules and the 190 test molecules, and to describe how the SCScore-based stratified sampling was performed.

- **The curriculum design lacks thorough justification.** The two-stage split at SCScore = 4 is presented without extensive motivation. Why is 4 the right threshold? How sensitive is performance to this choice? An ablation varying the SCScore cutoff or the ratio of Stage 1 to Stage 2 training steps would be informative.

- **Missing details on the "Terminate" action.** The agent can issue a TERMINATE action, but the conditions under which this is learned or optimal are not well discussed. Does the agent learn to terminate early when it detects a solved route, or does it sometimes terminate prematurely?

- **Presentation issues.**
  - The paper contains several typos (e.g., "traed" for "treated", "belif" for "belief" in Section 4.2; "retrosynthsis" in Section 1; "pecialized" in Section 5.1).
  - The term "overlapped reaction" used in Figures A.3 and A.4 is never defined in the text.
  - The relationship between the discount factor γ used in Equation (8) and the training dynamics could be better explained.

---

> ### Author Rebuttal · Authors · 2026-03-31
>
> We sincerely thank Reviewer NWE3 for the thoughtful feedback. Below we respond point by point. Additional rebuttal figures/tables are at https://anonymous.4open.science/r/RetrOrchestratorRebuttal-FCC3/ and cited as Fig./Table rX.
>
> **[W1] Comment on reproducibility**
> - **Code and checkpoints will be released under MIT upon acceptance.**
>
> **[W2/Q2] Comment on benchmark scale**
> - To address this concern, we also evaluate on a larger **PDB-600** benchmark (Table r5).
> - On PDB-600, RetrOrchestrator remains above the reported static planners in Table r5 (**68.83%** vs **53.83%** and **43.67%**).
> - On Retro*-190, our **95% CI is [90.53%, 97.26%]**, so the lower bound matches the **90.53%** reported in [1](No McNemar test due to result availabilit).
>
> **[W3/Q1] Comparison to ensemble**
> - For the ensemble baseline, one "expansion" means **one joint call to all 7 SSRs**.
> - We therefore updated the Pareto analysis to compare **total SSR calls**, not raw expansion count (Fig. r1).
> - Under this fair accounting, the ensemble is **no longer Pareto-optimal**, while RetrOrchestrator remains on the Pareto front.
>
> **[W4] Comparison with recent method**
> - Thank you for pointing out PDVN. PDVN is reported with **50 predictions per SSR call**, while our main setting maxed at **10** for all methods.
> - Under the same 10-prediction setting, PDVN+Retro* achieves **47.89%**.
> - With the budget of 50, ours reaches **96.84%**, above the reported **93.68%** for PDVN+Retro*.
> - Table r6 and Fig. r1 in the link summarize the success-cost trade-off.
>
> **[W5] Wallclock time profiling**
> - Inference time splits into **19.3% LLM**, **69.2% SSR calls**, and **11.5% environment overhead** (Fig. r4).
> - Among SSR calls, **Chemformer (43.3%)** and **RootAligned (27.2%)** dominate runtime.
>
> **[W6/Q3] Failure case analysis**
> - We find four main failure modes:
> - **Search budget**: **5/11** are solved with larger expansion; **1** remains unsolved by all methods.
> - **Higher complexity**: failures have higher sp3 (**0.60 vs 0.46**), more stereocenters (**3.91 vs 1.83**), and more aliphatic rings (**1.91 vs 1.56**).
> - **Rare motifs**: silyl (**18.2% vs 5.3%**) and organotin (**9.1% vs 0.5%**) are enriched.
> - **Weak SSR support**: max similarity to USPTO training data is only **0.31-0.78**; **10/11** have no neighbor >=0.7.
> - Overall, the failures mainly reflect **finite search budget + sparse SSR coverage**.
>
> **[W7] Theoretical Contribution**
> - We respectfully clarify that our contribution here is **not** that an LLM can generally serve as a POMDP policy.
> - Rather, the contribution is the **retrosynthesis-specific POMDP reformulation** in which SSR selection becomes part of the action space, together with **scaffold-aware credit assignment and curriculum learning**.
>
> **[W8] Comment on data leakage**
> - Training molecules come only from the **Retro* training split**; there is **no overlap** with Retro*-190 test molecules.
> - We provide the distribution plot (Fig. r5) and will provide the training dataset if allowed.
> - For SCScore stratified sampling, we bin by SCScore and sample uniformly across bins.
>
> **[W9/Q4] Ablating the curriculum design**
> - We agree this ablation is important. Under the same **300-step** budget (Table r3), the **two-stage curriculum is critical**: Stage-1-only **65.43%**, Stage-2-only **72.22%**, full training **94.21%**.
> - Changing the SCScore cutoff to **3.5 / 4.5** also hurts (**74.26% / 72.11%**).
> - We choose **4.0** because Stage 1 must provide a **useful KL reference** for Stage 2: **3.5** is too weak, while **4.5** is too restrictive.
> - So the gain comes from a **functional two-stage optimization**, not arbitrary tuning.
>
> **[W10] Details of the "TERMINATE" action**
> - **TERMINATE is rewarded only for a solved route; otherwise it is penalized.**
> - We do not observe premature termination at evaluation time.
> - This is consistent with our response to Reviewer **Gnxp [Q1]** on why stopping is part of the learned action space.
>
> **[W11] Presentation issues**
> - We appreciate the careful reading and will correct the typos in the camera-ready.
> - In Fig. A.3/A.4, "overlapped reaction" means **identical reaction**, used to highlight later-stage switching from EditRetro to Graph2Edits / Chemformer.
>
> **[Q5] Performance drop on PDB-120**
> - This is consistent with prior OOD findings (RetroOOD [2]): PDB ligands are substantially more out-of-distribution.
> - Residual failures are enriched in **high-sp3, stereochemically complex, rare motifs**, with weak SSR support.
> - We therefore believe the drop reflects **OOD chemistry + limited SSR coverage**, not instability of our method.
> - **Most methods degrade under this regime; RetrOrchestrator still performs best.**
>
> We thank the reviewer again and would be happy to clarify further if helpful.
>
> [1] Song, Xiaozhuang, et al. "AOT*: Efficient Synthesis Planning via LLM-Empowered AND-OR Tree Search."
>
> [2] Yu, Yemin, et al. "RetroOOD: Understanding out-of-distribution generalization in retrosynthesis prediction."

---

> > ### Author Rebuttal · Reviewer_NWE3 · 2026-04-02
> >
> > Thanks to the authors for their constructive rebuttal. The additional experiments and analyses directly address several of my main concerns and meaningfully strengthen the paper.
> >
> > While the code release promise (W1) is appreciated, reproducibility remains unverifiable at this stage. This is still a clear weakness of the study.

---

### Official Review · Reviewer_GBfP · 2026-03-13

**Soundness:** 3
**Presentation:** 3
**Significance:** 2
**Originality:** 3
**Overall Recommendation:** 4
**Confidence:** 3

**Summary:**

This paper presents RetrOrchestrator, a multi-step retrosynthesis framework that reformulates the problem from a traditional MDP into a POMDP. The framework uses an LLM agent to jointly decide which molecule to expand and which single-step retrosynthesis model to call during search. To support this, the authors treat heterogeneous single-step models as callable tools and introduce scaffold-aware GRPO together with a two-stage curriculum training strategy based on molecular difficulty.

**Compliance With Llm Reviewing Policy:**

Affirmed.

**Final Justification:**

The author's response addressed my concerns. But I still think:

1. Using a router to select SSR models has been studied before, which limits the novelty of this work.

2.  It is still difficult to assess how well the approach would generalize to larger retrosynthesis benchmarks or more diverse chemical spaces, as it only evaluates on Retro*-190.

So I would be neutral on this paper.

**Key Questions For Authors:**

1. Could you analyze the learned policy over the “number of predictions to generate” action dimension?

Since the action space jointly includes which molecule to expand, which SSR model to use, and how many reactions to generate, it would be helpful to understand whether the agent learns a meaningful adaptive branching-factor policy rather than only a routing policy over SSRs. For example, does it request more predictions on harder or more uncertain intermediates, and how much of the final gain seems attributable to this action component?



2. Could you provide a sensitivity analysis for the two-stage curriculum design?

The current training setup separates Stage 1 and Stage 2 using SCScore-based difficulty and specific data sizes for easy vs. hard molecules. I would be interested in knowing how sensitive the final result is to the SCScore threshold, the stage ordering, and the easy/hard sampling ratio. This would help clarify whether the reported gains depend on a fairly specific curriculum design or reflect a more robust training recipe.


3. Could you provide a more chemistry-facing interpretation of the learned routing policy?

The paper already shows cluster-level correlations and one qualitative example of successful late-stage switching, which is useful, but I would like to better understand what chemical knowledge the agent has actually internalized. For instance, are there recurring scaffold motifs, reaction types, or molecular contexts for which particular SSRs are consistently preferred, and can these patterns be summarized in a way that would be informative to chemists beyond the aggregate correlation plots?

**Limitations:**

yes

**Strengths And Weaknesses:**

Strengths:

1. Instead of only improving search over a fixed one-step model, the paper makes model choice itself part of the planning policy, which clearly differentiates it from classical MCTS-style planners and Retro*. That is a reframing for multi-step retrosynthesis, especially given the paper’s evidence that different SSRs perform differently across scaffold clusters.


2. The expanded action space, POMDP formulation, and scaffold-aware advantage estimation are all motivated by the same core issue: sparse delayed rewards and state-dependent SSR quality. The scaffold-based grouping is a thoughtful chemistry-specific adaptation.


3. On Retro*-190, the final system reaches a reported 94.21% success rate, outperforming strong baselines and also exceeding the strongest reported specialized LLM baseline in the table.

Weaknesses

1. The method relies heavily on the assumption that different SSR models have durable complementary strengths. This is plausible today, but the paper’s own analysis shows substantial convergence toward EditRetro, suggesting that a generally stronger SSR could reduce much of the value of dynamic orchestration. The result stems from model heterogeneity instead of a better planning principle.


2. The paper under-utilizes the LLM. Although the method is framed as an LLM retrosynthesis agent, in practice, the LLM mainly acts as a router/controller over existing SSR tools, rather than directly proposing or refining promising retrosynthetic transformations. Relative to recent LLM retrosynthesis work that uses the LLMs more in pathway generation or route design, this feels like a conservative use of the LLM’s capabilities [1,2].

3. It remains unclear whether the full agentic RL/POMDP machinery is actually necessary. The experiments show that dynamic orchestration helps, but they do not isolate whether most of the gain comes from the LLM-based planner or simply from learning a good state-dependent SSR router.


4. The evidence for genuinely dynamic multi-model behavior is not fully convincing. The paper provides one qualitative example of beneficial switching, but the aggregate analysis still suggests a broad drift toward EditRetro. I think a more direct statistic showing how often success truly depends on using multiple SSRs, rather than mostly defaulting to one very strong model, is needed.

5. The evaluation is narrow for the breadth of the paper’s claims. The main evidence is on Retro*-190, with an additional OOD result on a 120-molecule PDB subset. This is limited to arguing that dynamic orchestration is a generally superior planning paradigm.


[1] Chang, Liao, et al. "How Well Can LLMs Synthesize Molecules? An LLM-Powered Framework for Multi-Step Retrosynthesis."

[2] Liu, Wei, et al. "Retro-R1: LLM-based Agentic Retrosynthesis." The Thirty-ninth Annual Conference on Neural Information Processing Systems. 2025.

---

> ### Author Rebuttal · Authors · 2026-03-31
>
> We sincerely thank Reviewer GBfP for the thoughtful feedback. Below we respond point by point. Additional rebuttal figures/tables are in https://anonymous.4open.science/r/RetrOrchestratorRebuttal-FCC3/ and cited as Fig./Table rX.
>
> **[W1] Model heterogeneity over planning principle**
> - We agree model heterogeneity is central here. **No SSR is universally optimal.** Fig. 1 shows clear cluster-dependent strengths across models.
> - Even the best static planner (EditRetro) is **13.68 points** below ours (**94.21% vs 80.53%**).
> - The policy also **does not simply collapse to EditRetro**: **92.48%** of successful routes use multiple SSRs (Fig. r2), and Fig. A.3/A.4 shows cases where switching SSRs succeeds while EditRetro alone fails.
> - We therefore view **explicitly planning over transition-model heterogeneity** as part of the planning principle itself. A static planner assumes one transition model is sufficient everywhere; our results suggest that assumption is unnecessarily restrictive.
>
> **[W2] LLM under-utilization**
> - We respectfully view the LLM as being used where it adds the most value: **state-dependent coordination across heterogeneous SSRs**, while chemistry prediction stays within specialized SSR models.
> - With only a **1.5B** LLM, RetrOrchestrator reaches **94.21%**, above **77.21%** in [2].
> - This is also more parameter-efficient than **236B** DeepSeek-V2.5 [1] and a **7B** fine-tuned LLM [2].
> - As SSRs improve, this division of labor becomes **more scalable**, not less.
>
> **[W3] Comparison against state-dependent SSR router**
> - We appreciate this suggestion and trained a strong non-LLM state-dependent router using the current molecule, search-graph features, and model identity (Table r7).
> - It achieves **82.63%**, still well below our **94.21%**; the gap is significant (**McNemar p=0.0003**).
> - This rules out the hypothesis that the gain comes from a stronger state-dependent router alone, as **a dedicated non-LLM router still falls significantly behind**.
>
> **[W4] Statistics of SSRs used**
> - Only **7.51%** of successful routes use a single SSR, while **92.49%** use two or more (Fig. r2).
> - This suggests success comes from **combining complementary SSRs**, rather than defaulting to one strong model.
>
> **[W5] Evaluation scale**
> - We appreciate this concern. The same trend also holds on larger PDB evaluation, as discussed in our response to Reviewer **P1bx [W3/Q2]**.
>
> **[Q1] Learned policy over "number of predictions to generate"**
> - We agree this deserves separate analysis (Table r8). The policy learns an **adaptive branching factor**, not just SSR routing.
> - **Search-stage dependence**: average `no_pred` increases over search steps (**$ \rho=0.2$**).
> - **Chemical dependence**: `no_pred` correlates with descriptors / motifs such as Hall-Kier alpha (**0.297**), piperidine (**0.274**), TPSA (**-0.347**), lactam (**-0.398**), and pyridine (**-0.315**).
> - It is **not explained by model identity alone**: the direct association with queried SSR is weak (**$\eta^2=0.017$**), and feature effects vary across models.
> - Overall, the learned branching policy is **adaptive and chemically interpretable**.
> - We interpret this as the agent requesting broader expansion on later, harder surviving states, rather than using a fixed branching factor throughout search.
>
> **[Q2] Sensitivity analysis of the curriculum design**
> - We appreciate this question. We analyzed this in **Table r3**, with the fuller curriculum ablation discussed in our response to Reviewer **NWE3 [W9/Q4]**.
> - The **two-stage curriculum is critical**: Stage-1-only and Stage-2-only reach **65.43%** and **72.22%**, versus **94.21%** for full training.
> - Changing the SCScore cutoff to **3.5 / 4.5** also hurts (**74.26% / 72.11%**), supporting **4.0** as the best trade-off because Stage 1 must provide a useful KL reference for Stage 2.
>
> **[Q3] Chemistry-facing interpretations**
> - We agree this aspect should be clearer to chemists, so we added a **motif-conditioned analysis** (Table r9).
> - **EditRetro** is favored in stereochemical / heterocycle-rich settings, including **piperidine-, chiral-, and amide-containing** molecules.
> - **Chemformer** is favored in **aromatic / heteroatom-rich** settings, including benzene-rich, high-TPSA, and pyridine-containing molecules.
> - **LocalRetro** shows a smaller but consistent preference for **fused-ring** contexts.
> - These are **specialization patterns, not one-expert-per-motif**: action-level preference and molecule-level dominance do not perfectly coincide.
> - Overall, we believe the policy learns **chemically meaningful specialization**, not generic model preference.
>
> We thank the reviewer again and would be happy to clarify further if helpful.
>
> [1] Chang, Liao, et al. "How Well Can LLMs Synthesize Molecules? An LLM-Powered Framework for Multi-Step Retrosynthesis."
> [2] Liu, Wei, et al. "Retro-R1: LLM-based Agentic Retrosynthesis." The Thirty-ninth Annual Conference on Neural Information Processing Systems. 2025.

---

> > ### Author Rebuttal · Reviewer_GBfP · 2026-04-04
> >
> > Thanks for your response. I have adjusted my score accordingly.

---

> > > ### Author Response · Authors · 2026-04-08
> > >
> > > Thank you for the thoughtful feedback. We would like to briefly clarify two points from the final justification.
> > >
> > > **Using a router to select SSR models has been studied before, which limits the novelty of this work**
> > > > We respectfully clarify that, to the best of our knowledge, prior work has **not** clearly studied a **learned, state-dependent policy** that dynamically selects among multiple SSR models **during multi-step retrosynthesis planning**.
> > > > - The concern may stem from conflating several **related but distinct** settings:
> > > >     - **fixed-SSR substitution within planners** [1],
> > > >     - **benchmarking planner–SSR combinations** [2],
> > > >     - **single-step ensembling or multi-model system support** [3,4].
> > > >   - These are relevant baselines and contextualize our setting, but they do not study the same problem as ours.
> > > > - More importantly, our method is **more than a router**:
> > > >   - it is an **agentic planning policy** that makes **sequential decisions during search**,
> > > >   - it decides not only **which SSR to invoke**, but also **which molecule to expand** and **when to terminate**,
> > > >   - and it is optimized for **long-horizon planning success**, rather than only local routing quality.
> > >
> > > **It is still difficult to assess how well the approach would generalize to larger retrosynthesis benchmarks or more diverse chemical spaces, as it only evaluates on Retro\*-190**
> > > >We also respectfully clarify that the current evaluation is **not limited to Retro\*-190**.
> > > > - The paper already includes **OOD evaluation on the PDB dataset**, reported in **Table A.1** of the appendix and further summarized in **Table r5** of the rebuttal.
> > > >  - These results were included precisely to assess robustness **beyond the standard benchmark**, rather than only within Retro*-190.
> > > >
> > > > While broader large-scale evaluation across more diverse chemical spaces would certainly strengthen the picture and remains valuable future work, we believe the current submission already provides **meaningful evaluations beyond Retro\*-190 alone**.
> > >
> > > We thank the reviewer again for the careful review and hope these clarifications help with the final assessment.
> > >
> > > [1] Torren-Peraire, P. et al. *Models Matter: the impact of single-step retrosynthesis on synthesis planning.* Digital Discovery, 2024.
> > >
> > > [2] Choe, J. et al. *Retrosynthetic crosstalk between single-step reaction and multi-step planning.* Journal of Cheminformatics, 2025.
> > >
> > > [3] Maziarz, K., et al. *Chemist-aligned retrosynthesis by ensembling diverse inductive bias models.* NeurIPS 2025 AI for Science Workshop.
> > >
> > > [4] Tu, Z. et al. *ASKCOS: an open-source software suite for synthesis planning.* Accounts of Chemical Research, 2025.

---

### Official Review · Reviewer_P1bx · 2026-03-13

**Soundness:** 3
**Presentation:** 3
**Significance:** 3
**Originality:** 3
**Overall Recommendation:** 4
**Confidence:** 4

**Summary:**

This paper studies multi-step retrosynthesis planning and proposes RetrOrchestrator, an LLM-based agent that dynamically selects among multiple single-step retrosynthesis models (SSR) during planning. The authors argue that existing methods typically rely on a fixed SSR model, while different models exhibit varying performance across molecular scaffolds. To address this issue, the paper formulates retrosynthesis planning as a POMDP and treats each SSR model as a tool that the agent can invoke.
The proposed system uses a Qwen2.5-1.5B LLM as the planning policy and trains it with reinforcement learning. The authors introduce Scaffold-Aware GRPO (SA-GRPO), a variant of GRPO that incorporates scaffold-based clustering to derive step-level advantages and improve credit assignment in long-horizon planning tasks. The training also employs a two-stage curriculum to separate chemical rule learning from strategic planning.
Experiments on the Retro-190 benchmark* show that the proposed method achieves a 94.21% success rate, outperforming both traditional search-based planners and prior LLM-based planners, while achieving a favorable trade-off between success rate and computational cost.

**Compliance With Llm Reviewing Policy:**

Affirmed.

**Final Justification:**

The paper proposes a learning-based orchestration framework for retrosynthesis planning with strong empirical results. The rebuttal provides useful clarifications, particularly showing that the gains are not solely due to heuristic combinations, which increases my confidence in the learned policy. It also offers additional analysis of computational cost. However, the method remains relatively expensive, and the overall contribution, while meaningful, is somewhat incremental. Overall, I find the work solid and potentially useful, and I maintain my score.

**Key Questions For Authors:**

1. How sensitive is the proposed SA-GRPO method to the scaffold clustering threshold and similarity metric?
2. The experiments mainly focus on the Retro*-190 benchmark. Could the authors comment on how the method scales to larger retrosynthesis datasets or industrial-scale synthesis planning scenarios?
3. Since the framework relies on multiple SSR models as tools, how does performance change when the number of available models is reduced or increased?

**Limitations:**

1. The system depends on the availability of multiple pretrained SSR models.
2. RL training may require substantial computational resources.
3. The approach relies on scaffold similarity heuristics, which may not capture deeper reaction chemistry.

**Strengths And Weaknesses:**

Strengths:
1. The paper identifies an important limitation of existing retrosynthesis planning systems, namely that most approaches rely on a fixed single-step retrosynthesis (SSR) model throughout the planning process. The empirical observation that different SSR models perform unevenly across molecular scaffolds provides a clear motivation for dynamically selecting models during search.
2. The proposed framework introduces an interesting agent-based perspective for retrosynthesis planning. By formulating the planning problem as a POMDP and treating SSR models as tools callable by an LLM agent, the method enables joint decision-making over molecule expansion and model selection during search.
3. The scaffold-aware variant of GRPO is a reasonable attempt to address the credit assignment challenge in long-horizon planning tasks. Leveraging scaffold similarity to cluster intermediate states provides a domain-informed way to construct more informative learning signals.
4. The experimental results on the Retro*-190 benchmark are strong. The reported success rate of 94.21% exceeds both traditional search-based planners and several LLM-based baselines, and the paper includes ablations and analyses that help illustrate how the learned routing policy behaves.

Weaknesses:
1. The methodological novelty appears somewhat limited. Many core components of the framework, including LLM-based tool-calling agents, GRPO-based reinforcement learning, and POMDP-style formulations, build upon existing ideas. The main contribution lies in integrating these components for retrosynthesis planning rather than introducing fundamentally new algorithms.
2. The proposed scaffold-aware advantage estimation is intuitively motivated but lacks theoretical justification. It remains unclear why scaffold clustering should systematically improve credit assignment compared with alternative grouping strategies.
3. The experimental evaluation is mainly conducted on the Retro*-190 benchmark, which is relatively small. It is therefore difficult to assess how well the approach would generalize to larger retrosynthesis benchmarks or more diverse chemical spaces.
4. The overall system introduces additional complexity by combining an LLM planner, reinforcement learning training, and multiple SSR models. While the paper discusses inference efficiency in terms of model calls, it does not analyze overall training cost and computational requirements in depth.
5. Some implementation details that may affect reproducibility are not fully specified in the main paper, such as prompt structure, clustering threshold selection, and sensitivity to key hyperparameters.

---

> ### Author Rebuttal · Authors · 2026-03-31
>
> We sincerely thank Reviewer P1bx for the thoughtful feedback. Below we respond point by point. Additional rebuttal figures/tables are in https://anonymous.4open.science/r/RetrOrchestratorRebuttal-FCC3/ and cited as Fig./Table rX.
>
> **[W1] Comment on algorithmic novelty**
> - We respectfully clarify the main point most relevant here: the method changes the planning problem into a **joint policy over molecule choice, SSR choice, branching factor, and stopping**, rather than a fixed-SSR search procedure.
> - The gain is also not explained by any one borrowed component in isolation: Table r7 shows a strong non-LLM router remains below our method, while Tables r3-r4 show that both the curriculum and the mixed episodic/step-level advantages are necessary.
> - More clarifications on our contributions can be found in our response to **Reviewer Gnxp [W1].**
>
> **[W2/Q1] Ablating grouping strategy and similarity threshold**
> - We agree this ablation is important. A good grouping strategy must be **dense enough to avoid sparse rewards** but **specific enough to stay chemically meaningful**.
> - This is also the motivation in Sec. 4.3 of the submitted paper: exact molecule matching creates a sparsity problem with about **20% singleton groups**, while our chosen scaffold grouping at **$\delta=0.95$** reduces the singleton rate to about **6%** and yields a mean cluster size of **7.43**.
> - Tables r1-r2 show scaffold-level grouping with **0.95** is best at **94.21%**. Looser thresholds hurt (**0.9: 90.63%, 0.8: 71.58%**), and a stricter one also hurts (**0.999: 73.68%**).
> - Molecule-level grouping is too specific (**74.21%**). Search-graph-level grouping is competitive but lower (**88.42%**).
> - So performance is driven by **signal granularity, not scaffold identity**. Scaffold-level + **0.95** is our best interpretable choice.
> - We think the search-graph result is also informative: it shows the gain is **not tied to scaffolds per se**, but to **construct a learning signal with the right granularity**.
>
> **[W3/Q2] Relatively small benchmark**
> - We agree Retro*-190 is relatively small, so we also report OOD results on **PDB-120** and a larger **PDB-600** subset.
> - On PDB-600 we obtain **68.83%**, consistent with PDB-120 (**65.83%**), so the gain from dynamic orchestration is **not confined to the original subset**.
> - Detailed larger-scale comparisons and failure patterns are also provided in our responses to Reviewer **NWE3 [W2/Q2]** and **[W6/Q3]**.
>
> **[W4] Training cost and computational requirements**
> - We appreciate this practical question. One full run takes about **1 week on 4xA100 GPUs**.
> - Figure r3 shows rollout collection accounts for **62.4%** of training time, while scaffold clustering is only **0.1%**.
> - Figure r4 provides the rollout breakdown.
> - The detailed inference-time decomposition is shown in **Fig. r4** and discussed in our response to Reviewer **NWE3 [W5]**.
>
> **[W5] Reproducibility**
> - We appreciate the request for more detail.
> - **Prompt structure**: Appendix I, Lines 938-1103.
> - **Threshold analysis**: Appendix D.
> - The camera-ready will include rebuttal ablations on **threshold, curriculum, grouping strategy, and advantage weighting**.
> - **Code and checkpoints** will be released under MIT upon acceptance.
> - We also provide the rebuttal tables for these hyperparameter studies directly in the link (**Tables r1-r4**), so the main training choices can already be inspected quantitatively.
>
> **[Q3] Impact of SSR tool sets**
> - We do not believe **more SSRs alone explain the gain**.
> - The best single SSR (EditRetro) and the static all-SSR ensemble both remain below our method.
> - The improvement comes from **state-dependent selection over complementary models** with different strengths across molecule states, not simply a larger tool set.
> - Fuller multi-SSR usage statistics are shown in **Fig. r2** and discussed in our response to Reviewer **GBfP [W4]**.
> - In other words, additional SSRs help only when they contribute **non-redundant, complementary capabilities** that the policy can exploit adaptively.
> - We are conducting SSR pool-size / composition ablations and will update them in the link.
>
> We thank the reviewer again and would be happy to clarify further if helpful.

---

> > ### Author Rebuttal · Reviewer_P1bx · 2026-04-05
> >
> > The rebuttal clarifies several points, including the joint policy formulation and additional results on larger datasets, which help address part of my concerns. However, it is still not entirely clear how much of the gain comes from policy learning versus strong heuristic combinations, and the computational cost remains relatively high.

---

> > > ### Author Response · Authors · 2026-04-08
> > >
> > > We thank Reviewer P1bx for the thoughtful follow-up. We address the two remaining concerns below.
> > >
> > > **How much of the gain comes from policy learning versus strong heuristic combinations?**
> > > > We respectfully clarify that the gain cannot be attributed solely to heuristic combination. Our empirical results validate that it arise from **the full learned policy**,  which exploits richer search-state information during planning.
> > > >  - To better isolate the contribution of **policy learning**, we additionally trained a **non-LLM model router** on trajectories collected during SA-GRPO training, as noted in our response to Reviewer GBfP [W3] (Table r7).
> > > >    - Under the same setting with molecules selected by Retro* and the maximum prediction budget, this learned router achieves **82.63%** success rate.
> > > >    - While this is already stronger than all static baselines, it still remains substantially below **RetrOrchestrator (94.21%)**.
> > > > - This is also consistent with our analysis in Reviewer GBfP [Q1–Q3], where the correlations between a range of heuristic signals and action choices are only **moderate** (Tables r8–r10).
> > > >   - Since the non-LLM router is trained on these heuristic-style features, the remaining performance gap suggests that the improvement is not explained by heuristic combination alone, but by the **full learned policy**.
> > >
> > > **Why is the computational cost still relatively high?**
> > > >We respectfully clarify that the additional runtime is justified in two senses:
> > > >- it reflects a favorable **effectiveness–efficiency trade-off** in this setting, rather than overhead introduced by our method
> > > >   - Our claim is **not** that RetrOrchestrator is uniformly cheaper than traditional fixed-SSR planners.
> > > >   - Rather, our point is that it achieves a substantially better **effectiveness–efficiency trade-off**, as shown in Fig. 6.
> > > >   - To make this transparent, we added more detailed cost analysis in the rebuttal, including **runtime breakdown** (Fig. r3) and **training profiling** (Fig. r4).
> > > >     - These analyses show that most of the overhead comes from **SSR calls**, rather than the LLM itself.
> > > >     - This is important because SSR inference is the dominant cost of the **heterogeneous models**, not a bottleneck introduced specifically by our agent design.
> > > >
> > > >- the overall **LLM-related cost remains lower than prior LLM-based retrosynthesis planners**.
> > > >   - Our method uses a **1.5B** open-weight model trained on **4×A100 40G GPUs** for roughly one week.
> > > >   - By comparison, prior LLM-based planners have used **commercial LLMs with hundreds of billions of parameters** [1–3], or a **7B** model trained on **32×A800 80G GPUs for 70 hours**, which also reported about **20 minutes per molecule** during inference [4].
> > > >
> > > >Thus, while RetrOrchestrator does introduce additional cost relative to traditional planners, the **LLM component itself remains comparatively lightweight** relative to recent LLM-based alternatives.
> > >
> > >
> > > We hope these additional results help clarify the remaining concerns, and we would be grateful if the reviewer could take these additional analyses into account in the final assessment.
> > >
> > > [1] Chang, Liao, et al. *How Well Can LLMs Synthesize Molecules? An LLM-Powered Framework for Multi-Step Retrosynthesis.*
> > >
> > > [2] Wang, Haorui, et al. *LLM-Augmented Chemical Synthesis and Design Decision Programs.* ICML, 2025.
> > >
> > > [3] Song, Xiaozhuang, et al. *AOT*: Efficient Synthesis Planning via LLM-Empowered AND-OR Tree Search.*
> > >
> > > [4] Liu, Wei, et al. *Retro-R1: LLM-based Agentic Retrosynthesis.* NeurIPS, 2025.

---

### Decision · Program_Chairs · 2026-04-30

**Decision:**

Accept (regular)

**Comment:**

This paper proposes RetrOrchestrator – an LLM-based retrosynthesis agent that jointly selects which next molecule to expand and which single-step model to use. Authors show competitive results on the Retro*-190 benchmark with still reasonable wall-clock runtimes compared to baseline non-LLM-based approaches.

On the positive side, reviewers considered the work to be reasonably presented, with the POMDP formulation being generally principled and well-motivated. They also noted strong results on Retro*-190 (during rebuttal, comparison to one recent method was raised, which reported better results, but when setting is exactly matched RetrOrchestrator still wins). One reviewer also pointed to the LLM agent showing emergent behaviour that is interesting in its own right.

On the negative side, some reviewers pointed to somewhat limited algorithmic novelty, and discussed computational demands. While the addition of the LLM agent does present an additional overhead, authors argued the relatively small size of that model, combined with how it is used, make the overhead manageable; the paper does report computational requirements and shows that RetrOrchestrator is fast in the Pareto sense (e.g. when one fixes the target solve rate and sees how long it takes to achieve that).

In the end, all reviewers were in favour of accepting this paper, also praising the strong rebuttal, which further strengthened the work. After reviewing all information, I lean into the reviewer consensus, and also recommend acceptance.